# Rate-induced tipping in ecosystems and climate: the role of unstable states, basin boundaries and transient dynamics

Ulrike Feudel[1]

[1]Institute for Chemistry and Biology for the Marine Environment, Carl von Ossietzky University Oldenburg, Oldenburg, Germany

**Correspondence:** Ulrike Feudel (ulrike.feudel@uni-oldenburg.de)

**Abstract.** The climate system as well as ecosystems might undergo relatively sudden qualitative changes in the dynamics when environmental parameters or external forcings vary due to anthropogenic influences. The study of these qualitative changes, called tipping phenomena, requires the development of new methodological approaches that allow modeling, analyzing, and predicting observed phenomena in nature, especially concerning the climate crisis and its consequences. Here we briefly
review the mechanisms of classical tipping phenomena and investigate in more detail rate-dependent tipping phenomena which occur in *non-autonomous systems* characterized by multiple timescales. We focus on the mechanism of rate-induced tipping caused by basin boundary crossings. We unravel the mechanism of this transition and analyze, in particular, the role of such basin boundary crossings in non-autonomous systems when a parameter drift induces a saddle-node bifurcation in which new attractors and saddle points emerge, including their basins of attraction. Furthermore, we study the detectability of those
bifurcations by monitoring single trajectories in state space and find that depending on the rate of environmental parameter drift, such saddle-node bifurcations might be *masked* or *hidden* and they can be detected only if a critical rate of environmental drift is crossed. This analysis reveals that unstable states of saddle type are the organizing centers of the global dynamics in non-autonomous multistable systems and as such need much more attention in future studies.

## 1  Introduction

The climate system consists of many interacting components (Ghil and Lucarini, 2020; Franzke et al., 2015). These components could either be different compartments of the climate system itself, like e.g., atmosphere, hydrosphere, cryosphere, and biosphere, or, zooming into one of such compartments, e.g., the velocity components of the ocean flow or the abundances of different species in an ecosystem. Though there is no standard definition of a complex system, many researchers agree on the following properties inherent to such systems: (1) The interactions between the components and/or the external forcing of
the system are, in general, characterized by nonlinearities that give rise to various positive and negative feedbacks possibly resulting in unexpected changes of the dynamics when intrinsic parameters or external forcings are varied. (2) These nonlinear interactions lead to the emergence of a remarkably complex, partly unpredictable temporal dynamics or to the ability of the system to spontaneously form temporal, spatial or spatiotemporal patterns. (3) Random fluctuations, which are unavoidable in natural systems, lead to dynamics that are governed by both deterministic and random behaviors.

The study of the impact of nonlinearities in the geosciences has a long history concerning investigations of (i) chaotic dynamics leading to obstructions to predictability (Lorenz, 1963; Smith et al., 1999; Tel et al., 2020), (ii) scaling properties of various geophysical processes (Lovejoy and Schertzer, 2012; Schertzer and Lovejoy, 2011), (iii) the formation of coherent structures in flows (Wiggins, 2005; Mancho et al., 2013; Haller, 2015; d'Ovidio et al., 2004) and their impact on marine ecology (Kai et al., 2009; Rossi et al., 2014; Sandulescu et al., 2007) to name only a few. Several phenomena in the climate
system, like the breakdown of the Atlantic part of the thermohaline ocean circulation (Atlantic Meridional Overturning Circulation – AMOC) (Rahmstorf, 1995; Weijer et al., 2019; Lohmann et al., 2021), the loss of Arctic sea ice (Notz, 2009; Eisenman and Wettlaufer, 2009; Eisenman, 2012), or the loss of species in ecosystems (Gossner et al., 2016; Binzer et al., 2012; Ficetola and Denoel, 2009) have been discussed in the past in terms of classical bifurcation theory. i.e., the approach describing qualitative changes in the long-term dynamics of a nonlinear system when a control parameter is varied and crosses
critical thresholds (Guckenheimer and Holmes, 1986; Ott, 1992; Alligood et al., 1992). The study of such transitions from a more general point of view has been intensified during the last decade due to the need to develop an appropriate mathematical methodology to tackle the problems of the climate crisis. Two properties of the climate system call for extensions of classical bifurcation theory leading to the new notion of *critical transitions*: (1) Most processes are *multi-scale processes* in space and time, i.e., different physical, chemical or biological processes immanent to the system evolve on different temporal and spatial
scales. (2) Many critical transitions are observed in a time-dependent environment, such that the control parameters – either intrinsic to the dynamics or to the strength of an external forcing – vary with a certain arbitrary time-dependence manifested by a specific trend. As long as the environment evolves on a much slower timescale than the intrinsic dynamics, classical bifurcation theory is still appropriate, and a quasi-stationary approach can be used to study the response of a nonlinear system concerning climate change. This situation changes as the rate of environmental change becomes comparable with the timescale of the in-
trinsic dynamics, particularly the rate of dissipation in the system (Kaszás et al., 2016). While classical bifurcation theory has been developed to deal with models that are either autonomous or periodically forced systems, the most important challenge is now to extend the notion of critical transitions to non-autonomous systems. In addition, temporal changes in the external forcing following specific trends and mimicking climate change often happen on a different timescale than that of the intrinsic dynamics of the system under consideration. These timescale separations lead to partly unexpected behaviors. Such scale
dependence has been studied in the literature in different contexts such as e.g. climate sensitivity (Bastiaansen et al., 2022), tipping in excitable systems (Pierini and Ghil, 2021; Vanselow et al., 2022), overshooting and reversing tippings (Ritchie et al., 2021, 2023) and the topological structure of invariant sets in complex systems including their characteristics like fractal dimensions (Alberti et al., 2023; Charo et al., 2021).

In this paper, we review the classification of tipping phenomena, explain their mechanisms, and give examples of their
occurrence in climate science and ecology (Sec. 2 and Sec. 2.1). Special emphasis is given to discussing the impact of the different timescales of the various physical, chemical, or biological processes. Therefore, the main focus is on rate-induced transitions in which the rate of change in environmental conditions leads to new bifurcation phenomena in non-autonomous systems. We explain the different consequences of rate-induced transitions and point out that time-dependent variations of an external forcing can shift the focus from stable long-term states (attractors) to unstable states of saddle type and their stable and

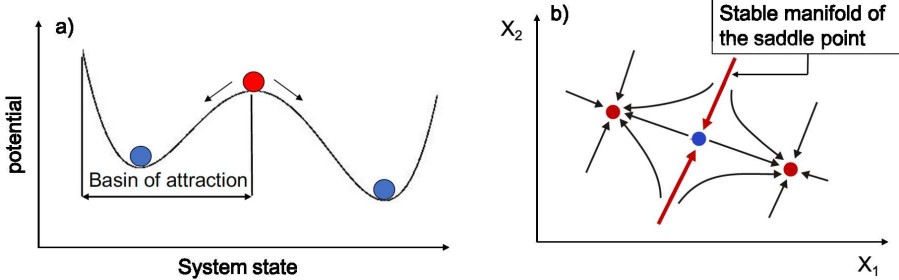

**Figure 1.** (a) Sketch of the potential (stability landscape) of a bistable system and (b) state space of a bistable system including the unstable saddle point and its stable manifold which makes up the boundary between the two basins of attraction.

unstable manifolds (Sec. 3). This approach leads to a fresh view on the role of unstable states in the dynamics of any nonlinear system subject to a parameter drift. In addition, we show that classical bifurcations, which occur in a nonlinear system due to time-dependent changes of internal parameters or external forcing, can be masked for certain initial conditions depending on the rate of parameter change. This masking is due to time-dependent changes in the basins of attraction, which might happen in the course of the variation of external forcing. More specifically, the boundaries of the basins of attraction, which separate regions of qualitatively different behavior in state space, start moving under time-dependent variation of intrinsic parameters or external forcing, giving rise to unexpected changes in the course of certain trajectories. This process leads to a variation of the relative size of the basins of attraction and to rate-induced tipping of trajectories by "crossing basin boundaries". Finally, we discuss the results in Sec. 4 and point out several consequences of this dynamics such as, e.g., the occurrence of other than rate-induced critical transitions can occur without any warning.

## 2 Multistability as a prerequisite for tipping phenomena

In many cases, tipping phenomena require the simultaneous existence of several different stable states of a system under the same given environmental conditions. Bi- and multistability can best be illustrated by a stability landscape represented as a potential (Fig. 1a). The stable states are in the valleys, while the unstable states are on the hills. This view of a stability landscape in terms of a potential is always correct for a one-dimensional system following a dynamical equation $\dot{x} = f(x) = -\frac{dV}{dx}$ or in case of a higher-dimensional gradient system represented by $\dot{\mathbf{x}} = \mathbf{f}(\mathbf{x}) = -\nabla V$. However, most nonlinear dynamical systems are not gradient systems, but even in that case, one can compute the stability landscape as a quasipotential (Freidlin and Wentzell, 1998; Graham et al., 1991; Cameron, 2012).

Here, stability of a state means linear stability with respect to small perturbations given by the eigenvalues of the corresponding Jacobian matrix for ,e.g., steady states $x^{(s)}$ $J_{ij} = \frac{\partial f_i}{\partial x_j}|_{x=x^{(s)}}$, or the Jacobian matrix $\mathbf{J_p}$ of the Poincaré map for periodic states. Small perturbations are damped out by the system due to dissipation, and the system returns to its original state. To view it in the picture of the stability landscape: the disturbance displaces the stable state visualized by the blue ball away from the valley and "rolls" back again due to the steepness of the walls of the potential $V$ corresponding to the strength of the restoring

forces. Directly on the hill, which relates to an unstable state, only a very small disturbance is needed to initiate the ball to "roll" either to one or the other side of the hill, depending on the direction of the disturbance. The unstable state (red ball) located on

the hill of the stability landscape marks the basin boundary. This boundary separates the two basins of attraction, i.e., the two sets of initial conditions which all converge to one of the respective attractors. In higher dimensional systems these unstable states on the boundary are of saddle type, possessing stable and unstable manifolds. The stable manifolds are hypersurfaces in state space whose dimension is equal to the number of stable directions or stable eigenvalues of the corresponding Jacobian matrix of the saddle, while the unstable manifolds correspond to hypersurfaces determined by the number of unstable direc-

tions or eigenvalues. In the special case of a two-dimensional system the saddle steady state has two eigenvalues one stable and one unstable and the corresponding stable and unstable manifolds are one-dimensional. This is illustrated in Fig. 1b). The stable manifolds along which trajectories move towards the saddle make up the basin boundaries.

In this setup, critical transitions are associated with a relatively sudden qualitative change of the dynamics in which the system moves from one stable state to another, i.e., the system tips by getting from one valley into the other by different

mechanisms. In general, those mechanisms are related to certain disturbances, kicking the system out of the position in the valley such that the other valley can be reached. However, it is essential to note that a specific tipping phenomenon, namely rate-induced tipping, does not necessarily require the existence of bi- or multistability. Instead, in those critical transitions, it is sufficient that the system trajectory moves into a part of the state space with different properties. Therefore tipping, in general, cannot always be identified with the well-known classical bifurcations but can, particularly in rate-induced tipping, only be

explained as bifurcations in non-autonomous systems.

## 2.1 Mechanisms of tipping and the role of different timescales

Often the picture of the stability landscape mentioned above is translated into a specific bifurcation diagram exhibiting hysteresis, showing the two stable states and the unstable one separating those two depending on the intrinsic parameters of the system or the external forcing (cf. Fig.2). In the representation of the stability landscape, it is rather simple to explain the different

disturbances that cause a system to tip. On the one hand, the state variables such as, e.g., temperature and salinity as quantities determining the water density in the ocean or the abundance of species in an ecological system can be disturbed. These disturbances correspond to the displacement of the state from the valley of the fixed stability landscape, depicted as a vertical path of perturbations $ds_i$ in Fig. 2. On the other hand, disturbances in the system parameters or external forcings change the stability landscape, corresponding to a horizontal path of perturbations $dp_i$ in Fig. 2. Both types of disturbances are possible and have

very different effects (Schoenmakers and Feudel, 2021). Such disturbances can occur in three different ways: (1) fluctuations, i.e., small random perturbations of the state variables or the driving forces of the system that satisfy certain statistics, (2) large individual disturbances that correspond to extreme events or shocks, and (3) changes in environmental conditions or driving forces associated with certain trends, whereby a rate of change can characterize this trend. In nature, one would always observe a combination of these disturbances, but for theoretical investigations of tipping mechanisms, it is helpful to analyze the

individual types of disturbances separately.

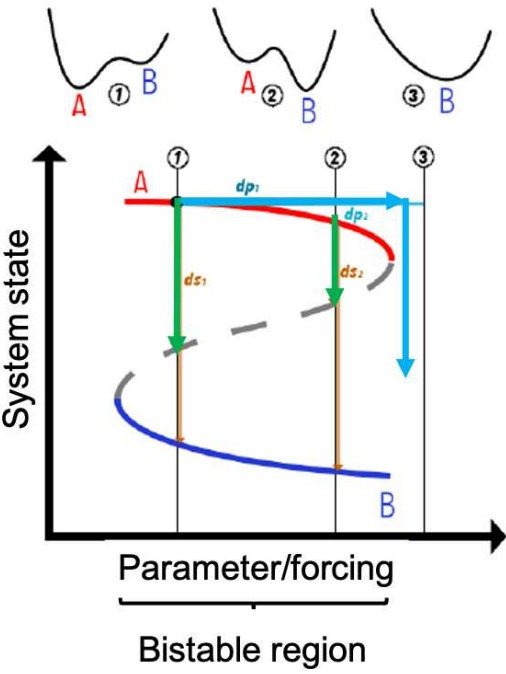

**Figure 2.** Relationship between the stability landscape and the control parameters/the forcing of the system. Cyan arrows indicate the changes in control parameters or forcings, the light green and light brown arrows refer to changes in the state variables.

Next, we illustrate the different tipping mechanisms in Fig. 3 following essentially the classification introduced by Ashwin et al. (Ashwin et al., 2012). To this end we look at a bistable system, as shown in the stability landscape in Fig. 1 and its disturbances in Fig. 2. Changing environmental conditions, represented as a control parameter, cause the stability landscape to change, and there is generally an interval of environmental conditions in which bistability is present. For illustration purposes, let us assume that the two stable states would be two alternative states in an ecosystem. The desired state denoted by $A$ is characterized by, e.g., a high biomass of plants and the lower undesired state $B$ is characterized by a low biomass of plants. Further, we assume that the increasing change of the environmental conditions points to rising habitat destruction either by land use change or climate change.

### 2.1.1 Bifurcation-induced tipping

Environmental changes, e.g., increasing habitat destruction, can affect the growth of plants. If those environmental changes are very slow, then the ecosystem state would slowly "move" along the red solid line $A$ to the right to smaller biomass states but still on the upper branch until it reaches the tipping point beyond which the state of high biomass $A$ ceases to exist. This tipping point is linked to the fact that one of the minima in the stability landscape disappears when the critical threshold value

of the environmental conditions is exceeded. Hence, the system tips into the low biomass state $B$, the only one existing for those environmental conditions. In general, such critical transitions, associated with a characteristic qualitative change in the stability landscape, such as the emergence of new or the disappearance of existing stable states, are called bifurcation-induced transitions (Fig. 3a).

Let us discuss some examples of bifurcation-induced tipping in the climate system and ecology. Over the past decade, a number of tipping elements, i.e., climate phenomena, have been identified as candidates expected to tip in the further course of climate change (Lenton et al., 2008; Schellnhuber et al., 2016; Armstrong McKay et al., 2022). Those tipping elements include ocean circulation, parts of the biosphere and the cryosphere. Specifically, these include, e.g., the AMOC, the Greenland ice sheet, the Arctic sea ice, and the Western Antarctic Ice Sheet as physical systems (hydrosphere and cryosphere), the Amazon rainforest, the tropical coral reefs and the boreal forests as ecological systems. In those systems, bi- or even multistability, i.e., the coexistence of more than 2 stable states for the same environmental conditions has been discovered. For the AMOC, often two different stable flow patterns exist: one of them can be considered as a conveyor belt transporting heat to the Northern latitudes, releasing this heat to the atmosphere, forming North Atlantic Deep Water (NADW) which is transported back to the Southern latitudes at considerable depth. This would be the state, where the heat transfer to the North is "on". The other stable state is related to an "off" state. This bistability can give rise to a possible breakdown of the AMOC, which has been discussed employing several conceptual models (Stommel, 1961; Rahmstorf, 1996; Rooth, 1982; Wood et al., 2019). In those conceptual models often the second state is related to a reverse circulation. In large ocean circulation models this bistability has also been confirmed (Weijer et al., 2012), with an "off" state which does not relate to a reverse circulation but to a very weak circulation northwards. In a large ocean circulation model it has been shown, that the system can exhibit the coexistence of several different flow patterns related to different spatial patterns of heat transfer to the atmosphere (Rahmstorf, 1995). This occurrence of multistability has been confirmed recently with other high-resolution models (Mehling et al., 2022; Lohmann et al., 2023).

The possible melting of the Arctic Sea ice is also discussed in terms of bistability comprising two stable states where in one of which the Arctic Sea ice disappears to a large extent in summer and shows ice cover only in winter (Notz, 2009; Eisenman and Wettlaufer, 2009; Eisenman, 2012), while the other corresponds to an ice cover for the whole year.

Examples of alternative states in ecosystems have been discussed in the literature (cf. Folke et al. (2004) and references therein), though the existence of thresholds in ecology is controversely debated (Hillebrand et al., 2020). A prominent example in which such transitions from one stable state to another has nowadays already been observed are tropical coral reefs, which are found to be overgrown with green algae due to climate change and other anthropogenic and non-anthropogenic influences. As a result, the system collapses and exhibits a shift from a coral-dominated into an algae-dominated reef (Holbrook et al., 2016). Another ecological example from Europe is shallow lakes, which, due to increasing nutrient inputs from agriculture, tip from a clear water state with high visibility at a large depth, which allows for plant cover at the bottom of the lake, to a turbid water state with high algae concentrations and no plants due to the lack of light for photosynthesis (Scheffer et al., 1993).

It is important to note that bifurcation-induced tipping is not restricted to the shown saddle-node bifurcation, but many other bifurcations, such as e.g. Hopf bifurcations, torus bifurcations and homoclinic bifurcations, can be related to tipping phenomena (Boettiger et al., 2013). There is a large variety of possible bifurcations in the mathematical literature, local

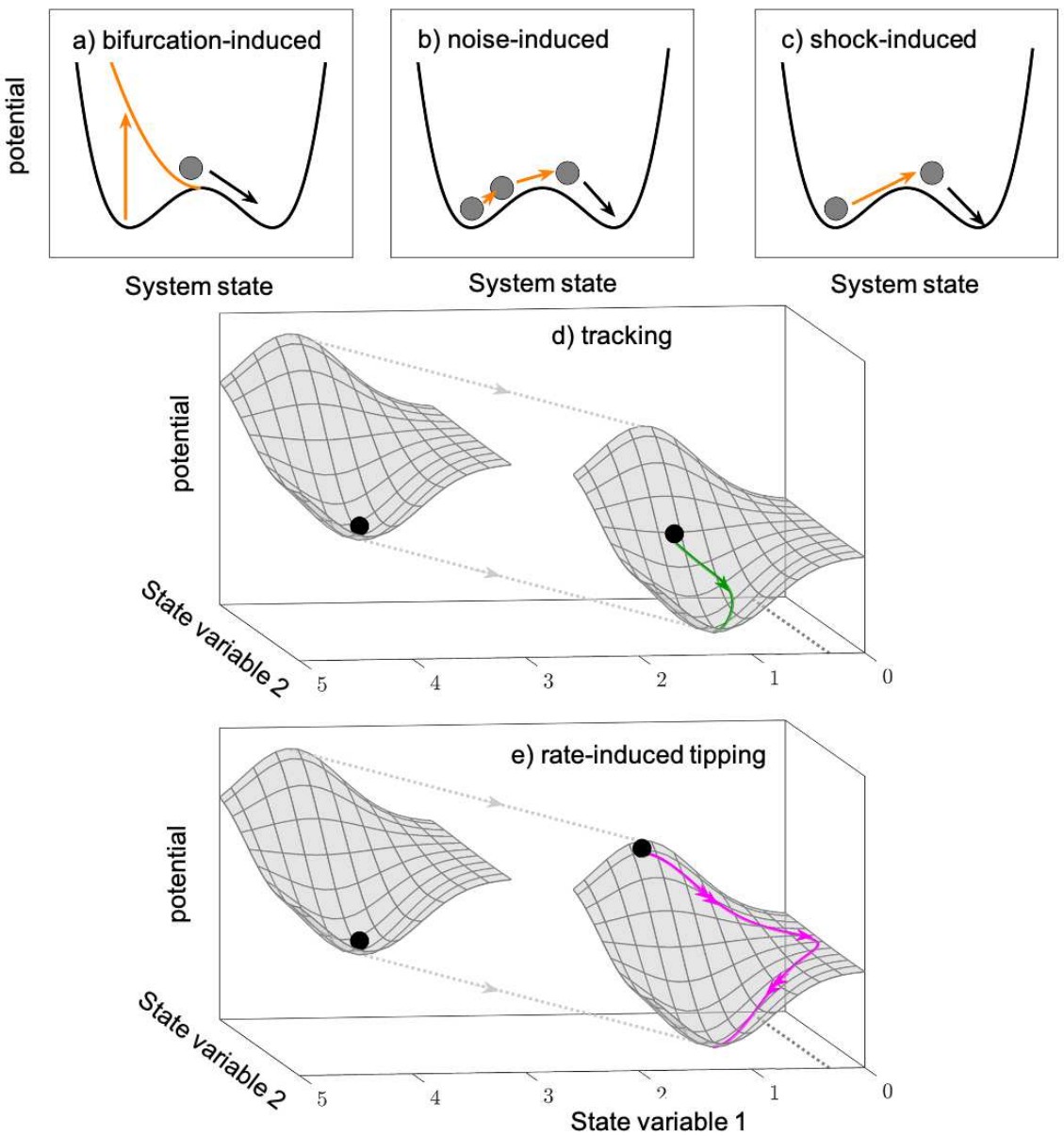

**Figure 3.** Illustration of the four different tipping mechanisms: a) bifurcation-induced tipping, b) noise-induced tipping, c) shock-induced tipping, d) and e) rate-induced tipping in the case where no alternatve states are present, but the rate-induced tipping refers to a very large excursion in state space with different properties. d) rate-induced tipping with a tracking trajectory, e) rate-induced tipping with a tipping trajectory. For more explanations, see the text.

(Guckenheimer and Holmes, 1986) and global (Kuznetsov, 1995) ones. Still, only a few of them have been addressed so far concerning tipping in climate and ecology.

### 2.1.2 Noise-induced tipping

This tipping process is caused by fluctuations (Fig. 3b) that cannot be avoided in natural systems, as all quantities describing the physical environment, such as, e.g., temperature and precipitation, are subject to fluctuations. Noise-induced tipping is based on the fact that the system, the ball, is permanently disturbed by the fluctuations of different sizes and directions at the minimum of the stability landscape. As a response to fluctuation-induced kicks of the ball away from the valley, it is "rolling" towards the minimum again because of the immediate onset of restoring forces. Since fluctuations occur at any instant, the next kick usually occurs before the original stable state is reached again. Therefore, a suitable sequence of kicks – the most probable exit path (Maier and Stein, 1992; Khovanov et al., 2008; Kraut and Feudel, 2003) – can push the system over the hill and, hence, it tips into the other state (Fig. 3b). In Fig. 2, this path would correspond to a sequence of disturbances that pushes the system's state along a vertical path $ds_i$ over the dashed line corresponding to the hill (green arrow) and subsequently approaching the alternative state $B$ along the brown arrow. Though this description sounds like a result of a single trajectory, it needs a stochastic description, since one has to study ensembles of trajectories with different realizations of the noise and probability distributions (pdfs) over the state space. There is a vast literature on noise-induced transitions in many different science disciplines and the notion of noise-induced transitions differs across the literature. While many studies classify noise-induced transitions as qualitative changes in the aforementioned pdf with the noise strength as the bifurcation parameter (Horsthemke and Lefever, 1984; Kuehn, 2011), other works focus directly on the transition from one stable state to another mentioned under the influence of noise as outlined above. Ashwin et al. (2012) define N-tipping as a system which leaves the neighborhood of a quasistationary state due to the influence of fluctuations.

Such noise-induced tipping is hypothesized to be responsible for the regime shift observed in the dominance of two species – a brittle star and a burrowing mud shrimp species – living in the sediment of the North Sea. This change in dominance took place at the end of the 1990s without any significant changes in environmental conditions and, hence, cannot be attributed to bifurcation-induced tipping but rather to a change in fluctuations in the water movement (Van Nes et al., 2007). This observation highlights an important property of noise-induced transitions: they can occur without any environmental changes, i.e., without changing the stability landscape. With a fixed stability landscape, the strength of the fluctuations, their statistical properties, and the height of the potential barrier that must be overcome are the main essential factors governing this transition.

Noise-induced transitions have also been shown to be a crucial mechanism of tipping in the climate system, as climate change involves not only shifting mean values such as global temperature associated with global warming (Freund et al., 2006) but also changing the variability of environmental parameters, such as, e.g., changing precipitation patterns in the Indian monsoon towards more extreme precipitation events (Goswami et al., 2006). Deep convection in the Labrador Sea as a part of deep water formation and heat exchange with the atmosphere in the AMOC was investigated as an example of noise-induced transitions in the climate system. This convection –a very local phenomenon– is bistable, i.e., there are years when deep convection events take place in so-called convection chimneys to drive deep water formation, and there are other years when this is not the case. It

has been shown that fluctuations in temperature and salinity in the ocean can contribute to a shutdown of convection chimneys and thus to a weakening of the AMOC (Kuhlbrodt et al., 2002; Lenderink and Haarsma, 1994).

### 2.1.3  Shock-induced tipping

While noise-induced tipping causes the system to tip through a whole sequence of small disturbances, shock-induced tipping is caused by a single large disturbance (Fig. 3c) which moves the system into the basin of attraction of another stable state. In nature, this could correspond to an extreme event that can push a system over the hill in the stability landscape. This tipping mechanism is closely related to the stability measure of ecological resilience introduced by Holling (1996), who considered the smallest possible disturbance that can cause a system to tip as the crucial determinant of resilience. In mathematical terms, this disturbance corresponds to the smallest distance to the basin boundary of a stable state, which is simple to compute in low-dimensional systems (Klinshov et al., 2015; Mitra et al., 2015), but needs to be calculated by an optimization procedure in high-dimensional systems (Halekotte and Feudel, 2020).

Calculating such smallest disturbances in ecological networks also provides valuable information about which system parts are most vulnerable to extreme disturbances. In this way, it can be shown that in networked ecosystems of plants and their pollinators, particularly those species have the highest extinction risk that are specialists or species that are part of a tree-like structure in the graph of the network having only a very loose connection to the core of the species' network (Halekotte and Feudel, 2020). Indications for the role of extreme events in tipping phenomena can also be found in coral reefs, where a massive decrease in sea urchins in an epidemic process has been identified as a crucial factor in the collapse of a coral reef. This transition can be interpreted as shock-induced tipping by the " epidemics " event combined with other extreme events such as two devastating cyclones (Mumby et al., 2007).

### 2.1.4  Rate-induced tipping

This tipping mechanism describes a system's response to an environmental change associated with a particular trend. It differs from those discussed so far by three essential points: (1) In this mechanism, the relationship between the timescales of the physical, chemical, and/or biological processes in the system under consideration, i.e., the intrinsic timescales and timescale or rate of the trend of environmental changes plays a decisive role. (2) This mechanism does not necessarily require the existence of alternative stable states. (3) The critical threshold value is not determined by a specific environmental parameter itself, but by the *rate of its change*, which is, of course, strictly speaking, also a parameter of the system, but a very particular one. In other words, the speed at which environmental changes occur is crucial. Additionally, it is important to note that this tipping phenomenon is associated with the dynamics in a *non-autonomous system*. For a thorough mathematical description, we refer to the seminal papers by Ashwin et al. Ashwin et al. (2012, 2017) and Wieczorek et al. Wieczorek et al. (2011).

While all aforementioned tipping mechanisms are related to the coexistence of alternative states, rate-induced tipping can also occur when there is only one stable state present and the system is characterized by different timescales (slow-fast system). The dynamics of such systems can be described by so-called critical manifolds in case of a perfect timescale separation or slow manifolds, when the timescale separation is finite. In case of a complex structure of the critical manifolds, for instance when

these critical manifolds have stable and unstable parts which meet in a fold, a rate-induced crossing of this fold can make the trajectory visit very different parts of the state space far away from the original stable state and perhaps even dangerous for the system (for a more mathematical description including the conditions under which this transition occurs see (Wieczorek et al., 2011). This mechanism of rate-induced critical transitions is illustrated in Fig. 3d,e where the whole stability landscape is moved at a certain rate. Suppose that the stability landscape in Fig. 3d is pulled with a certain rate towards the observer. Consequently, the ball will no longer be located in the minimum of the valley but will be displaced to the left. The restoring forces will start acting, and the ball begins to "roll" to catch the moving minimum. If the pulling rate is slow, then the stable state (ball) follows, or we say it *tracks* the minimum of the stability landscape. By contrast, in Fig. 3e, the rate of "pulling away" the stability landscape is much faster or comparable with the timescale of the restoring forces. In this case, the ball lands in a completely different region in state space, leaving the minimum's proximity and leading to qualitatively different dynamics. This large excursion in state space corresponds to rate-induced tipping since the system visits very different parts of the state space with qualitatively different behavior. If the change in the environmental conditions stops, this visit to a different state will be transient and, finally, the system returns to the stable quasi-stationary state, which has moved. This transient dynamics could lead to qualitatively different states, like population collapse in predator-prey systems (Vanselow et al., 2019) or population outbreaks (Vanselow et al., 2022). The second mechanism of rate-induced transitions occurs in multistable systems, where the trajectory can cross the basin boundary (Ashwin et al., 2012; O'Keeffe and Wieczorek, 2020; Lohmann et al., 2021), called basin instability by O'Keeffe and Wieczorek (2020), and basin crossing by Lohmann et al. (2021). We will discuss this mechanism of basin boundary crossing in the next section in more detail and omit here a sketch of it.

Again, let us look at an ecosystem as an example: If, for example, environmental changes occur very slowly, as in bifurcation-induced tipping, the species in the ecosystem have enough time to adapt to the changed environment. Conversely, if, e.g., climate change happens too fast, species adaptation fails, and, as a result, ecosystems can collapse. For example, this mechanism can be demonstrated in predator-prey systems, where the prey's habitat is destroyed by climate change or anthropogenic influences like land use change. It is possible to determine a critical rate of environmental changes beyond which the ecosystem collapses (Siteur et al., 2016; Vanselow et al., 2019). In other ecosystems, it can also happen that a particular species grows to very high abundances, for instance, forming a (possibly harmful) algal bloom (Vanselow et al., 2022). Such rate-induced critical transitions exist not only in ecosystems but also in physical systems, such as the Greenland ice sheet (Klose et al., 2023). In summary, considering that environmental destruction is nowadays accelerating, this tipping mechanism seems particularly dangerous.

### 2.1.5 Tipping in spatially extended pattern-forming systems

In the previous analysis, the considered systems were spatially homogeneous and, therefore, usually modelled by ODEs or time discrete systems (maps). However, complex systems in space are often characterized by the fact that they can spontaneously form spatially inhomogeneous patterns resulting from, e.g., a Turing bifurcation (Turing, 1952). With respect to tipping phenomena, pattern-forming systems play a special role since spatial interactions can lead to an acceleration or to a slowing down of tipping in adjacent points in space. Due to the spatial interactions, one observes often gradual tipping (Bel et al., 2012;

Siteur et al., 2014; Bastiaansen et al., 2020, 2022; Hasan et al., 2022), i.e., tipping is only visible locally. Therefore the tipping of the entire system is completed only on a much longer timescale: Instead of relatively abrupt critical transitions, one observes a transition that occurs "step by step" via different spatial patterns or by front propagation in systems possessing different coexisting spatial patterns. Several examples have been studied by Meron and coworkers in dryland vegetation models (Zelnik et al., 2013; Bel et al., 2012; Zelnik et al., 2018). These models of different complexity study the interplay between vegetation and soil water. Besides the homogeneous states "bare soil" and "full vegetation cover" there exist – depending on the environmental conditions, in general the precipitation level – different patterns like holes in the vegetation cover, stripes and spots of vegetation. These patterns can coexist and fronts separating the different pattern can occur. The speed of the fronts determines the speed with which one patterns is exchanged by the other leading to a gradual tipping between different patterns in the whole area.

## 2.2 Prediction of tipping points and early warning signals

In the course of climate change it becomes more and more important to find appropriate methods to predict tipping points and to identify early warning signals. One method that has been developed in the physics and chemistry literature is critical slowing down (CSD) of the restoring forces when a bifurcation-induced transition is approached (Heinrichs and Schneider, 1981; Ganapathisubramanian and Showalter, 1983; Tredicce et al., 2004; Scheffer et al., 2009). Resulting from these smaller restoring forces that bring the system back to its stable state after a perturbation, the response to inevitable noise is amplified, leading to a rising standard deviation (Surovyatkina, 2005; Carpenter and Brock, 2006) and an increasing lag-1 autocorrelation when approaching the bifurcation (Held and Kleinen, 2004; Dakos et al., 2008). These methods—critical slowing down and noise amplification—have become extremely popular over the last decade as possible early-warning signals. Besides those methods other statistical approaches have been developed to estimate how close we are to tipping points in the climate system and in ecology (Lenton, 2011; Lenton et al., 2012; Fan et al., 2021; Clarke et al., 2023) or how probable noise- and rate-induced transitions are (Ritchie and Sieber, 2017). They have been used to estimate the proximity of several tipping points in climate such as e.g. the melting of the Greenland ice sheet (Boers and Rypdal, 2021), the collapse of the Atlantic Meridional Overturning (Boulton et al., 2014; Boers, 2021) or the loss of the Amazon rainforest (Boulton et al., 2022). Despite these various applications of early warning signals they have also been critically discussed from various perspectives (Ditlevsen and Johnsen, 2010; Boettiger and Hastings, 2012; Wagner and Eisenman, 2015).

## 3 Basin boundary crossing in coupled bistable systems: the role of timescales

### 3.1 Rate-dependent basin boundary crossing in one-dimensional ecosystems

To analyze the role of timescales as well as the role of saddles (invariant sets of saddle type possessing stable and unstable manifolds), we will employ different simple models from population dynamics. We start with a one-dimensional model of the growth of a population influenced by an Allee effect. The Allee effect describes the ecological fact that certain populations

need a minimal critical population density to grow (Stephens et al., 1999), i.e., only with an initial density above the critical one the probability for successful reproduction is large enough to ensure the growth of the population. On the other hand, if the initial population density is below the critical one, the species goes extinct. In mathematical terms, this effect is included in the growth rate and can be written in the simplest form as follows:

$$\dot{X} = rX\left(1 - \frac{X}{K}\right)\left(b - X\right) - mX,$$ (1)

where $r$ is the growth rate, $K$ is the carrying capacity of the environment, $b$ is the minimal critical population density, and $m$ denotes the mortality rate. This model has three different steady states:

$$X^{(1)} = 0,$$ (2)

$$X^{(2,3)} = \frac{K+b}{2} \mp \sqrt{\frac{(K+b)^2}{4} - \left(bK + \frac{mK}{r}\right)}.$$ (3)

While the first and the third are stable, denoting either that the population gets extinct or reaches a population density close to its carrying capacity $K$ (if mortality $m$ is low), the second one is unstable and corresponds to a population density close to the minimal critical population density. Hence, we have a bistable system as long as the following condition is met: $0 < b < K$. The basins of attraction of the two different stable long-term states are separated by the unstable steady state. In this simple one-dimensional model, this unstable steady state is the only point making up the basin boundary.

Let us now assume that changes in the environment lead to changes in the critical population density $b$ with a certain rate $v$. This variation is assumed to happen on a finite time interval to ensure that the saddle-node bifurcation at which the unstable $X^{(2)}$ and the stable $X^{(3)}$ steady states merge is avoided. This excludes an extinction of the species due to bifurcation-induced tipping to $X^{(1)}$. The corresponding model system can be written as follows:

$$\dot{X} = rX\left(1 - \frac{X}{K}\right)\left(b - X\right) - mX,$$ (4)

$$\dot{b} = \begin{cases} v \text{ for } 0 \leq t \leq T_r, \\ 0 \text{ for } T_r < t \leq T_{end}. \end{cases}$$

We apply a linear drift of parameter $b$ in the interval $[b_{start}, b_{end}]$ with $b_{end} < K$. This drift extends over a time $T_r$ corresponding to a rate $v = (b_{end} - b_{start})/T_r$. For times larger than $T_r$, the parameter $b$ is held constant. Since we are interested in the dynamics in the whole state space, particularly in the dynamics of the basins of attraction and their boundaries, we always start with a set of initial conditions distributed on a regular grid and monitor the convergence of all initial conditions

and check their long-term dynamics. Furthermore, we always compare the dynamics to those without the parameter drift, i.e., the convergence to the final states in the *frozen-in case* with constant parameter $b = b_{start}$.

Varying the critical population density means moving the two quasi-stationary states $X^{(2)}(b(t))$ and $X^{(3)}(b(t))$, with $X^{(2)}$ corresponding to the location of the basin boundary. We expect that trajectories that have originally converged to $X^{(3)}$ will now tip and reach the population extinction $X^{(1)}$. By coloring the trajectories with two different colors indicating to which basin of

attraction they belong in the frozen-in case, we visualize how many trajectories now tip for the chosen rate of environmental change (Fig. 4).

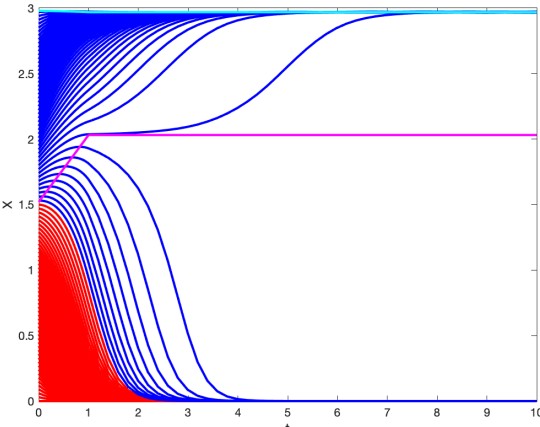

**Figure 4.** Time evolution of trajectories of system Eqs. (5) with parameter drift in $b$ with $1.5 \leq b \leq 2$. The movement of the saddle point $X^{(2)}$ demarking the basin boundary is shown in magenta, while the movement of the stable state $X^{(3)}$ is shown in cyan. All trajectories colored red would have converged to the extinction state $X^{(1)}$ in the frozen-in case $b = 1.5$, while blue trajectories would have converged to the high population state $X^{(3)}$. Parameters: $r = 2.0$, $K = 3.0$, $m = 0.02$, $v = 0.5$, $T_r = 1.0$.

We find that for this given rate of change of environmental conditions already, quite a large number of trajectories' initial conditions tip to extinction. Increasing the rate of change $v$ will force more and more trajectories to tip, and population extinction becomes more and more probable. The movement of the unstable steady state $X^{(2)}$ leads to an increase in the relative size of the basin of attraction of extinction $X^{(1)}$, while the relative size of the basin of attraction of the large population density $X^{(3)}$ shrinks. If we would allow $b(t)$ to increase even further, the basin of attraction of $X^{(3)}$ would finally disappear completely when the unstable saddle $X^{(2)}(b(t))$ merges with the stable large population density state $X^{(3)}(b(t))$ in a saddle-node bifurcation.

The mechanism, how trajectories in this one-dimensional case tip, is observable in Fig. 4, they cross the basin boundary directly when they "meet" the moving saddle point. Since the timescale with which the quasi-stationary saddle point moves and the speed of the trajectory are different, each initial condition possesses its own critical rate when it tips for the first time.

### 3.2 Basin boundary crossing and the emergence of multistability

To gain more insights into the interplay between the rate of moving basin boundaries due to environmental change and the intrinsic dynamics' timescale, we analyze a higher-dimensional problem in which smooth basin boundaries can be considered as hypersurfaces partitioning the state space into regions of different qualitative behavior. The basin boundaries correspond in the frozen-in case to the stable manifolds of a saddle point. For the sake of simplicity, we would like to analyze two coupled bistable systems, which can be coupled in two different ways, unidirectional and bidirectional. In the context of the simple population dynamical model analyzed above, it could be interpreted as two habitats (patches) bearing the same species which

can move or migrate between the habitats. An ecologically relevant bidirectional coupling would be migration based on the population differences between the habitats, i.e., a diffusive coupling.

To be more general, we choose to consider not only an ecological example but a general bistable model of the form:

$$\dot{X} = -\varepsilon(X - s_1)(X - s_2)(X - s_3). \tag{5}$$

Any bistable system can be brought into this form using a specific coordinate transformation outlined in (Kouvaris et al., 2012). The three steady states are given by $X^{(1)} = s_1$, $X^{(2)} = s_2$ and $X^{(3)} = s_3$ and depend in general on the intrinsic parameters of the system. The ecological example discussed above can be brought into that form by assuming a small input of species into each habitat resulting in two stable states related to high and low population densities, respectively, to avoid extinction (Sharma et al., 2015). Another example is the famous Schlögl reaction (Schlögl, 1972), an autocatalytic chemical reaction extensively studied since the 80s (Ebeling and Malchow, 1979; Grassberger, 1982; Mou et al., 1986).

Since our focus is on the role of timescales, we have introduced an additional parameter $\varepsilon$, which describes only the timescale of intrinsic dynamics of the system. For our ecosystem above, this would be the turnover time of the population, often denoted as the ratio between growth and mortality rate. To investigate the role of emerging and moving basin boundaries, we couple the two systems in three different coupling schemes: (1) unidirectional as a master-slave system, (2) bidirectional as a mutual forcing in both ways, and (3) bidirectional with a diffusion-like coupling. Though the diffusive coupling is the most relevant for many physical systems, we have included the other two coupling schemes here for more generality since they are widely used in the literature to investigate tipping cascades (Klose et al., 2020; Kroenke et al., 2020; Wunderling et al., 2021). The coupling strength determines how many stable states coexist in the coupled system. Taking the coupling strength as a bifurcation parameter in the classical sense as frozen-in, i.e., constant, the coupled system possesses one, two, three, or four attractors in the long-term limit. Changing the coupling strength in time means traversing the bifurcation diagram, as long as the rate of environmental change would be very slow such that the intrinsic dissipative timescale is fast enough to bring the system quickly to the attractor. However, we are interested in the case where intrinsic dynamics and the variation of the coupling strength are comparable in their timescales. In addition, we focus on the situation in which the number of stable long-term states changes and, with it, the global organization of the dynamics in state space. We consider the simplest case when two new steady states, an attractor and a saddle, are born in a saddle-node bifurcation, and a new basin of attraction emerges and grows. The stable manifolds of the saddle point emerging in such a classical saddle-node bifurcation make up the boundaries of the newly formed basin of attraction.

### 3.3 Coupled systems with master-slave coupling

As the first coupling scheme, we consider a unidirectional coupling corresponding to a master-slave configuration:

$$
\begin{aligned}
\dot{X}_1 &= -\varepsilon_1(X_1 - s_1)(X_1 - s_2)(X_1 - s_3) + CX_2, \\
\dot{X}_2 &= -\varepsilon_2(X_2 - s_1)(X_2 - s_2)(X_2 - s_3).
\end{aligned}
\tag{6}
$$

Rescaling the time in terms of $\tau = \varepsilon_1 t$, it turns out that only the ratios between the intrinsic timescales of the different subsystems $\varepsilon_2/\varepsilon_1$ and the ratios between the timescale of transport or coupling and the intrinsic timescale (like, e.g., $C/\varepsilon_1$) are important. Therefore we will continue the analysis with the rescaled equations:

$$
\begin{aligned}
\dot{X}_1 &= -(X_1 - s_1)(X_1 - s_2)(X_1 - s_3) + cX_2, \\
\dot{X}_2 &= -\varepsilon(X_2 - s_1)(X_2 - s_2)(X_2 - s_3),
\end{aligned}
\tag{7}
$$

where $\varepsilon = \varepsilon_2/\varepsilon_1$ and $c = C/\varepsilon_1$ are the corresponding ratios.

System 2 appears as a driver or master for system 1. We analyze the dynamics in the most intuitive way and use the concept of nullclines, which are given by the algebraic equations $\dot{X}_1 = f_1(X_1, X_2) = 0$ and $\dot{X}_2 = f_2(X_2) = 0$. While for the driver system 2, the nullclines are given by straight lines at the values of the three steady states of system 2, the nullcline of system 1 is represented by the cubic function $X_2 = (X_1 - s_1)(X_1 - s_2)(X_1 - s_3)/c$. The intersection points of $f_1 = 0$ and $f_2 = 0$ are the steady states of the master-slave system. Their stability can be computed from the eigenvalues of the corresponding Jacobian. An illustration of two possible situation is given in Fig. 5a,b. Depending on the internal parameters $\varepsilon$ and the coupling strength $c$, the system possesses two, three, or four stable, steady states in the considered parameter range of coupling strength $c$ ($c$ ranging from $0.4$ to $0.1$). The shown two cases serve as the beginning [Fig. 5a] and the end [Fig. 5b] point of the parameter drift along a linear ramp. We are fixing all parameters (frozen-in case) and compute the attractors and their corresponding basins of attraction by choosing a grid of initial conditions in a specified region of state space and integrating them all in parallel until they reach the attractor. This allows us to compute also the relative size of the basins of attraction $\mathcal{B}_\mathcal{A}$ as the quotient of the number of initial conditions converging to attractor $\mathcal{A}$ divided by the total number of initial conditions taken into account (Feudel et al., 1996). Figure 5 shows that the state space is "partitioned" into different basins of attraction indicated by different colors with basin boundaries separating them. In the frozen-in case, the basin boundaries are invariant sets that cannot be crossed by trajectories and, hence, represent rigid boundaries in state space for the trajectories. Throughout the paper, we fix the parameters defining the steady states of the uncoupled system to $s_1 = 1.0, s_2 = 2.0, s_3 = 3.0$ and vary only $c$ and $\varepsilon$ to focus on the role of timescales. This means the two systems are identical and vary only in their timescales and coupling strength, which, in this coupling scheme, corresponds to the strength of the impact of the driver.

To investigate the impact of a time-varying environment, we change the coupling parameter $c$ with a certain rate $v$. The corresponding dynamical system is now explicitly time-dependent. It involves a third differential equation for the dynamics of the environment, which is in the master-slave setup, the strength of the driver:

$$
\begin{aligned}
\dot{X}_1 &= -(X_1 - s_1)(X_1 - s_2)(X_1 - s_3) + cX_2, \\
\dot{X}_2 &= -\varepsilon(X_2 - s_1)(X_2 - s_2)(X_2 - s_3), \\
\dot{c} &= \begin{cases} v & \text{for } 0 \le t \le T_r, \\ 0 & \text{for } T_r < t \le T_{end}. \end{cases}
\end{aligned}
\tag{8}
$$

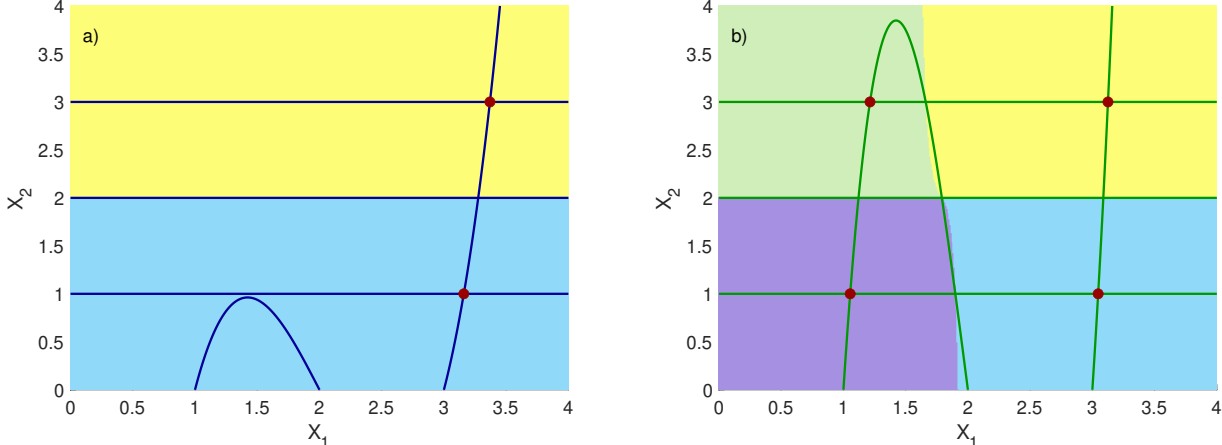

**Figure 5.** Basins of attraction, attractors and nullclines for the autonomous system Eqs. (8) with fixed coupling for two different constant coupling strengths $c$ (frozen-in case) and no timescale separation between system 1 and 2 ($\varepsilon = 1$): a) $c = 0.4$ and b) $c = 0.1$. Attractors are indicated by a dark red filled circle. Dark blue curves: nullclines $\dot{X}_1 = 0$ and $\dot{X}_2 = 0$ at $c = 0.4$, dark green curves: nullclines $\dot{X}_1 = 0$ and $\dot{X}_2 = 0$ at $c = 0.1$. The basins of attraction of the four attractors are plotted in light green, yellow, violet and light blue.

The most interesting dynamics happen with a parameter drift along which the number of attractors changes and saddle-node bifurcations lead to new attractors. This setup is suitable to elucidate the relative size of the basins of attraction and tipping probabilities for different initial conditions depending on the rate of change in the driver strength $v$.

     As mentioned above, classical bifurcations occur along the course of parameter variation giving rise to new invariant sets (in this case, steady states) and new basins of attractions, including their boundaries that lead to a new "partitioning" of the state

space with tremendous consequences for single trajectories. To illustrate this, we investigate the following scenario, which is inspired by the scenarios of parameter drift used by Kaszás et al. (2019): We change the impact of the driver decreasing $c$ with a constant rate $v$ in the interval $[0.4, 0.1]$ and as soon as $c = 0.1$ is reached it is kept constant. This change corresponds to a parameter drift or a ramping similar to many other papers studying rate-induced transitions (Ashwin et al., 2012; Vanselow et al., 2019). The advantage of this piecewise linear parameter drift is that we can compute the basins of attraction in the frozen-in

cases at the start at $t = 0$ and the end $t = T_{end}$ of the whole drifting process to have those frozen-in basins to quantify the change in the basins of attraction induced by the parameter drift.

     As in the previous subsection, we vary the rates of environmental change by varying the time interval $T_r$, keeping the $c$-interval fixed. In the course of this time evolution of $c$, the system is passing two saddle-node bifurcations. They can be computed analytically and happen at $c_{crit1} = 0.3849$ corresponding to $T_{sn1} = 0.1007$ and $c_{crit2} = 0.1283$ corresponding to

420 $T_{sn2} = 1.8113$, when the nullcline $X_2 = (X_1 - s_1)(X_1 - s_2)(X_1 - s_3)/c$ is touching the nullclines $X_2 = s_1$ and $X_2 = s_3$, respectively. The values given above for $T_{sn1}$ and $T_{sn2}$ are computed for the rate $v = -0.15$, i.e., $T_r = 2$. We start with the drift at $c = 0.4$ and end it at time $T_r$ at $c = 0.1$, corresponding to the two values for which we have shown the frozen-in basins

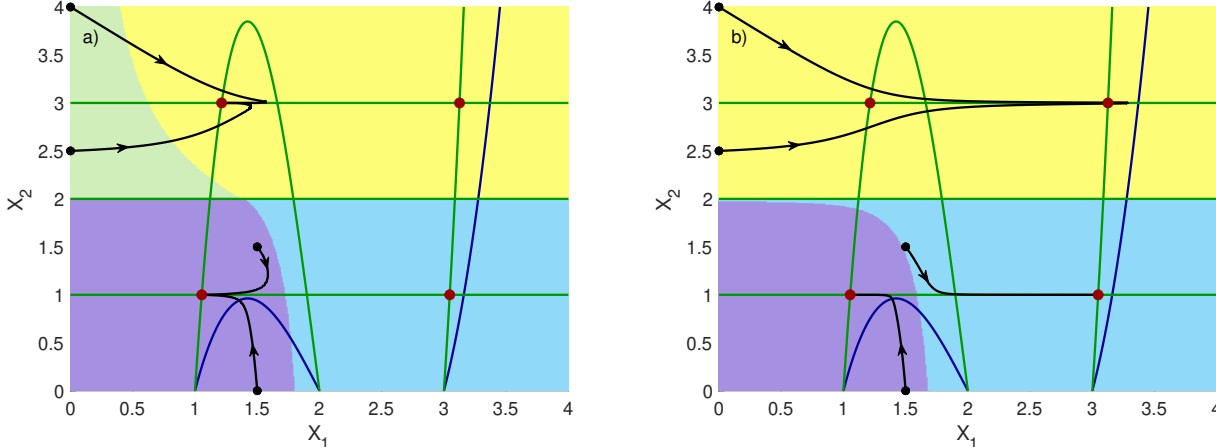

**Figure 6.** Time evolution of the non-autonomous system Eqs. (9) with decreasing impact of the driver $c$ in the interval $[0.4, 0.1]$ together with 4 exemplary trajectories ($\varepsilon = 1$): a) fast rate: $0 \leq t \leq 2.0$, $v = -0.15$ and $T_r = 2.0$ and b) slow rate with $0 \leq t \leq 8.0$, $v = -0.0375$ and $T_r = 8.0$. Attractors are indicated by dark red filled circles. Dark blue curves: nullclines at the beginning of the parameter drift at $t = 0$ and $c = 0.4$, dark green curves: nullclines at the end of the parameter drift at $t = T_{end} = 200$ and $c = 0.1$. The non-autonomous basins of attraction are presented at the end of the simulation $T_{end}$ in light green, yellow, violet and light blue.

of attraction in Fig. 5. Instead of a single trajectory, we again compute the time evolution of a whole grid of initial conditions to calculate the basins of attraction after the parameter drift and show additionally exemplarily 4 trajectories starting at different

positions in state space indicated by small black filled circles. Here we define the non-autonomous basin of attraction $\tilde{\mathcal{B}}(\tilde{\mathcal{A}})$ as the set of initial conditions which reach the quasistationary state $\tilde{\mathcal{A}}$ along a trajectory which includes the parameter drift. The result, shown in Fig. 6, demonstrates that the non-autonomous basins of attraction have been changed (compare to Fig.5). We note that the non-autonomous basins of attraction $\tilde{\mathcal{B}}$ look quite different from the ones of the frozen-in case $\mathcal{B}$, which indicates that the location of the boundaries, as well as the relative size of the basins of attraction, depend crucially on the rate of change

of the environmental forcing. New saddle points and their stable manifolds appear during the parameter drift. In the autonomous case, the stable manifolds would make up the boundaries of the newly formed basins of attraction. However, comparing the non-autonomous basin of attraction $\tilde{\mathcal{B}}$ at the final value $c = 0.1$ after the parameter drift and the frozen-in basin $\mathcal{B}$ reveals that the new boundaries of the basin of attraction after the drift are different from the stable manifolds of the corresponding saddle point as the saddle point does not even lie on the boundary of the non-autonomous basin of attraction $\tilde{\mathcal{B}}$.

To understand the tipping in more detail, we have plotted some particular trajectories and observed that some trajectories tip while others don't. While tipping trajectories change their course in state space when the bifurcation occurs to reach the newly emerging stable states, the tracking (non-tipping) trajectories follow their path largely undisturbed. In Fig. 6 (right panel), the drifting time $T_r = 8.0$ is relatively large corresponding to a low rate of change $v = -0.0375$ while in Fig. 6 (left panel) the drifting time $T_r = 2.0$ is relatively small corresponding to a larger rate of change $v = -0.15$. We note that for a fast drift,

more initial conditions reach the new stable states. This observation can be explained by the interplay between the dissipative timescale with which the trajectory is moving through the state space towards the quasi-stationary attractor and the rate of environmental change: if the trajectory is fast enough (for a slow environmental change), it will have converged already to a position close to the old attractor, when the new basin boundaries emerge. By contrast, when the environmental change is fast compared to the speed of the trajectory, the newly formed basin boundary will already have emerged, forcing the trajectory to change its course. As mentioned above, a tipping trajectory belongs at the start to one basin of attraction and changes on its course to another basin of attraction.

Looking at the basins of attraction as a whole, we note by comparing the right panel of Fig. 6 with the right panel of Fig. 5 that for the slow rate of environmental change, one of the attractors is not reached at all, the light green basin has "disappeared" at least in the region of state space considered. The reason for this behavior is twofold. One reason is the separation between the intrinsic timescale and the timescale of environmental change. The non-autonomous basins of attraction are computed starting from a grid of initial conditions, but all of them have already converged to the neighborhood of the attractor at $X_1 \simeq 3, X_2 = 3$, when the attractor ($X_1 \simeq 1, X_2 = 3$) appears in the saddle-node bifurcation due to a faster intrinsic timescale. On the other hand, the example studied describes either chemical concentrations or abundances of species, which have to be positive or equal to zero. This restriction of initial conditions are another reason, why none of the used initial conditions can reach the new attractor. They can only do so in the frozen-in case, but not beyond a critical rate of environmental change. As a consequence, the corresponding emerging attractor at ($X_1 \simeq 1, X_2 = 3$) would not have been observed for any of the considered trajectories. We say that this qualitative change in the state space resulting from the saddle-node bifurcation has been *masked* by the parameter drift. In other words, a global change in the dynamics would not have been noticed though we simulate a whole set of initial conditions covering the specified region in state space.

To study that further, we now look at the variation of the relative size of the basins of attraction for different rates of environmental change (Fig. 7a). We note that the relative size of the non-autonomous basins of attraction depends crucially on this rate of change $v$, and there is a critical rate $v_{crit}$ corresponding to a critical $T_{r_{crit}}$, where the light green basin is not visible anymore, corresponding to the masking effect. This rate-dependent transition in the relative basin size can be again explained by timescale arguments. For a slow rate of environmental change (large $T_r$), where the masking effect occurs, the dissipative timescale acts faster than the rate of change, i.e., the trajectory has already reached the neighborhood of the attractor before the bifurcation happens, i.e., before the qualitative change in the partitioning of the state space appears. This bifurcation is "hidden" for all the trajectories. By contrast, if the rate of change is very fast for smaller and smaller $T_r$, the bifurcation becomes "visible" since more and more trajectories change their course due to the new partitioning of the state space. This behavior is illustrated by the tipping probabilities indicating how many trajectories change from one basin to the other (Fig. 7b). We follow here the approach introdcued in (Kaszás et al., 2019) and compute the tipping probabilities $\mathcal{P}_{A_1 A_2}$. We calculate first the relative size of the non-autonomous basin of attraction $\tilde{\mathcal{B}}(\tilde{A}_2)$ for the quasisteady state $\tilde{A}_2$ at the end of the parameter drift simulation $T_{end}$ and take its intersection with the frozen-in basin of attraction $\mathcal{B}$ of the frozen-in attractor $\mathcal{A}_1$ at the beginning of the simulation normalized by the frozen-in basin of attraction $\mathcal{B}(\mathcal{A}_1)$ at the beginning of the simulation. Loosely speaking, we calculate that fraction of initial conditions which would have converged to the attractor $\mathcal{A}_1$ in the frozen-in case, but which

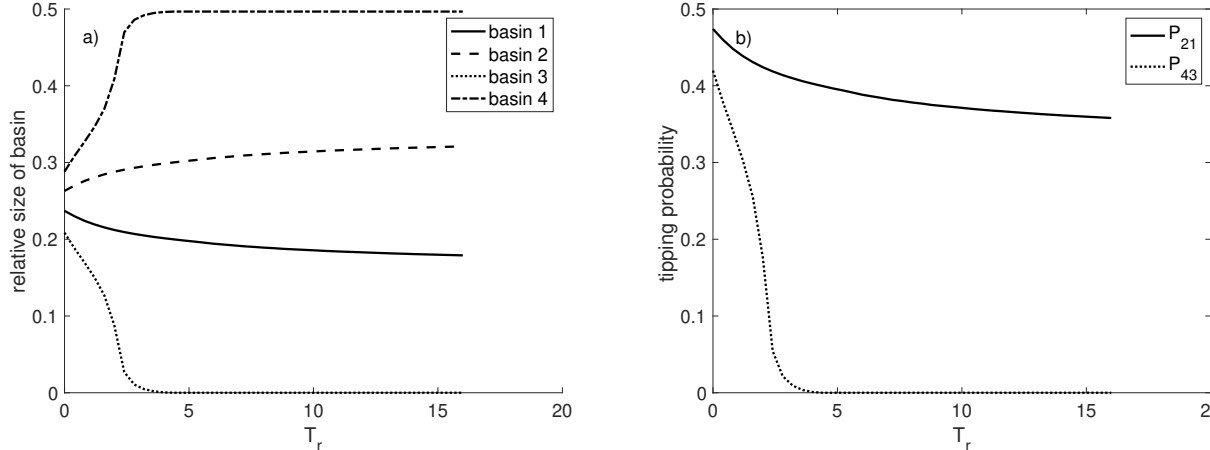

**Figure 7.** The relative size of the non-autonomous basins of attraction (a) and tipping probabilities (b) for the master-slave coupled system Eqs. (9) depending on the rate of environmental change measured in $T_r$, $\varepsilon = 1$, $c_{start} = 0.4$, $c_{end} = 0.1$ ).

during the parameter drift tip to the basin of attraction of the moving quasisteady state $\tilde{A}_2$. In mathematical terms this can be expressed as follows:

$$\mathcal{P}_{\tilde{\mathcal{A}}_1 \tilde{\mathcal{A}}_2} = \frac{\tilde{\mathcal{B}}(\tilde{A}_2) \bigcap \mathcal{B}(\mathcal{A}_1)}{\mathcal{B}(\mathcal{A}_1)} \tag{9}$$

The continuous change of the tipping probabilities shown in Fig. 7 indicates that we observe partial tipping, i.e. each trajectory possesses its own critical rate at which it tips.

To get deeper insights into the mechanism of basin boundary crossing of a particular trajectory, we analyse for $T_r = 2$ the trajectory starting at $(X_1, X_2) = (0, 4)$. It has been tipped when comparing Fig. 6a and b during the drift of $c$ from $c = 0.4$ and $c = 0.1$. The critical rate for this trajectory to tip for the first time is at $T_r = 2.1698$ corresponding to $v = -0.1383$. As illustrated in Fig. 8, the trajectory tips after the saddle-node bifurcation has happened, and the trajectory has reached the moving saddle point. The same applies to all other initial conditions on the non-autonomous basin boundary at this critical rate. Some additional example trajectories are shown in Fig. 8a,b, all leaving the basin of attraction via the saddle point. Some of them approach earlier the stable manifolds of the emerging saddle and move in its neighborhood towards the saddle quasistationary state. At exactly the critical rate, the trajectory will reach the saddle point along its stable manifold and stay there, but this cannot be demonstrated numerically because of the instability of the saddle point. This explains the long time intervals the trajectory spends in the vicinity of the saddle point before and after the tipping (Fig. 8c). At the moment of tipping all trajectories are crossing seemingly at the saddle point, which is part of the basin boundary of the frozen-in case, but not of the boundary of the non-autonomous basin. This means that basin boundaries in the non-autonomous case, according to our definition, are no longer identical with the moving stable manifolds of the corresponding saddle points but are made up by the set of initial

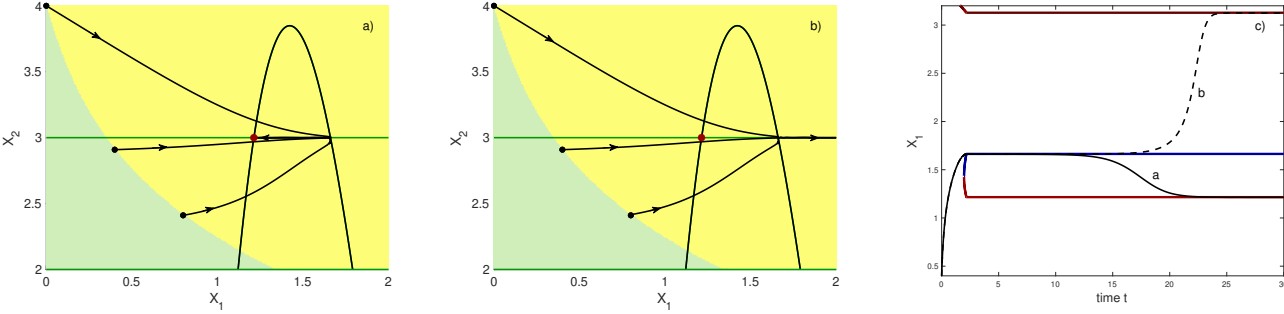

**Figure 8.** Tipping of trajectories starting next to the non-autonomous basin boundary in the points indicated by small black filled circles; a) just next to the boundary in the light green basin and b) just next to the boundary in the yellow basin. Black nullcline at the critical rate $T_r = 2.1698$. In this part of the state space only the light green and yellow non-autonomous basins are visible. c) time evolution of the two trajectories separated by the non-autonomous basin boundary with $X_1 = 0.4$ at $t = 0$ (middle trajectory in a) and b)).

conditions in the past which reach those moving stable manifolds after integration of the whole system including the parameter drift. These non-autonomous basin boundaries depend strongly on the rate of change of the environmental parameters.

So far, we have only varied the rate of environmental change but left the intrinsic timescales of the two systems equal to make the systems identical. However, different intrinsic timescales contribute also to a change in the dynamics. This approach is illustrated in Fig. 9 for the two different rates of the parameter drift. Now, one of the systems has a faster timescale than the other given by $\varepsilon = 0.1$. We find that, although bifurcations and nullclines remain the same, the non-autonomous basins of attraction change again, leading to rate-induced partial tipping of trajectories via basin boundary crossing. This finding

becomes visible by comparing Fig.6a and Fig. 9b and watching, e.g., the dynamics of the initial condition $(X_1 = 0, X_2 = 4)$.

Overall, we can conclude, that bifurcations which change the topological structure of the state space have a tremendous impact on the evolution of trajectories. This impact depends crucially on the relation between the intrinsic dissipative timescale and the timescale of environmental change. It turned out, that following the trajectories, which can be considered as observables does not necessarily detect those transitions. There is a detection limit beyond which bifurcations which happen in state space

are not noticed by the observables. As a consequence, transitions due to other tipping mechanisms like e.g. noise-induced tipping can happen without any warning.

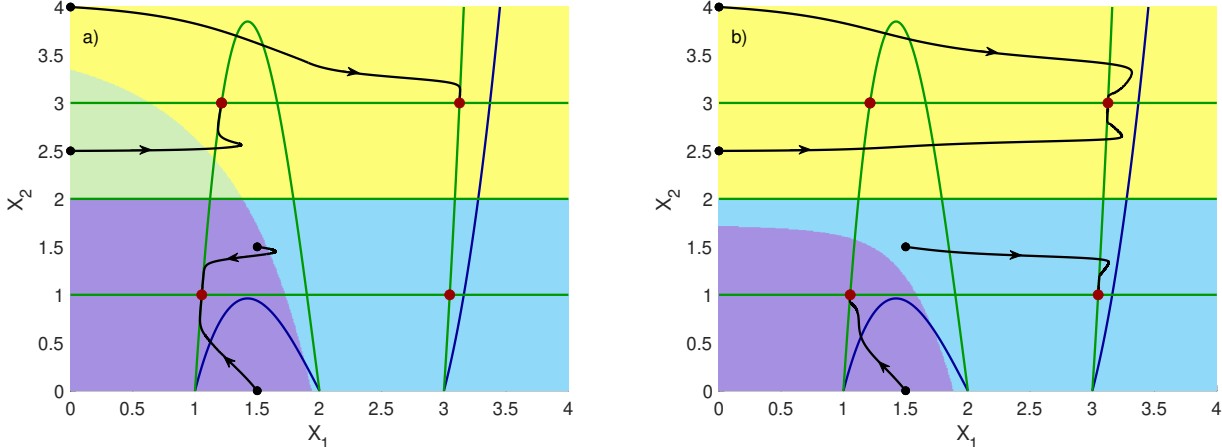

**Figure 9.** Time evolution of the non-autonomous system Eqs. (9) with decreasing impact of the driver $c$ in the interval $[0.4, 0.1]$ together with 4 exemplary trajectories (timescale separation between system 1 and 2 $\varepsilon = 0.1$): a) fast rate: $0 \leq t \leq 2.0$, $v = -0.15$ and $T_r = 2.0$ and b) slow rate with $0 \leq t \leq 8.0$, $v = -0.0375$ and $T_r = 8.0$. Attractors are indicated by dark red filled circles. Dark blue curves: nullclines at the beginning of the parameter drift at $t = 0$ and $c = 0.4$, dark green curves: nullclines at the end of the parameter drift at $t = T_{end} = 200$ and $c = 0.1$. The non-autonomous basins of attraction are presented at the end of the simulation $T_{end}$ in light green, yellow, violet and light blue.

### 3.4 Coupled systems with mutual forcing

The second type of coupling we consider is a mutual coupling of the two systems having different timescales and different strengths of impact on each other. This results in the following system of differential equations:

$$
\begin{aligned}
\dot{X}_1 &= -(X_1 - s_1)(X_1 - s_2)(X_1 - s_3) + c_1 X_2, \\
\dot{X}_2 &= -\varepsilon(X_2 - s_1)(X_2 - s_2)(X_2 - s_3) + c_2 X_1, \\
\dot{c}_{1,2} &= \begin{cases} v & \text{for } 0 \leq t \leq T_r, \\ 0 & \text{for } T_r < t \leq T_{end}. \end{cases}
\end{aligned}
\tag{10}
$$

Here the third differential equation applies either to $c_1$ or $c_2$. In contrast to the previous case of a master-slave coupling, the nullclines now vary with varying $\varepsilon$, leading to an even stronger impact of the timescale separation compared to the unidirec-
tional coupling. We study the tipping probabilities depending on the timescale separation and choose $\varepsilon = 0.1$ and extend the interval of varying the environment to $c_{1,2} \in [0.4, 0.01]$ to finally end up again with four attractors when the drifting process is finished.

Let us consider the symmetrical case in which we assume the same coupling $c_1 = c_2 = c$ and apply the variation to both parameters as indicated above. When checking again the dynamics in state space for two different rates of environmental change,
we note that the relative size of the basins changes even more dramatically not only depending on the rate of environmental

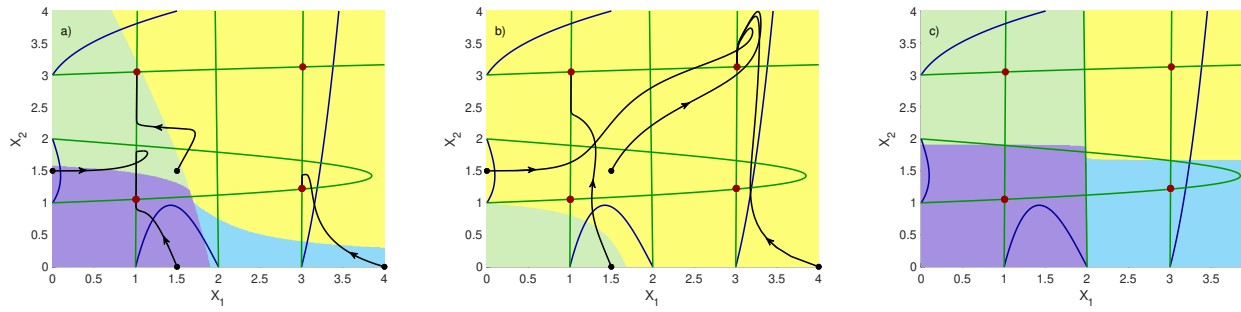

**Figure 10.** Time evolution of the non-autonomous system Eqs. (11) with decreasing impact of the driver $c$ in the interval $[0.4, 0.01]$ together with 4 exemplary trajectories (timescale separation between system 1 and 2 $\varepsilon = 0.1$): a) fast rate: $0 \leq t \leq 2.0$, $v = -0.15$ and $T_r = 2.0$ and b) slow rate with $0 \leq t \leq 8.0$, $v = -0.0375$ and $T_r = 8.0$. Attractors are indicated by dark red filled circles. Dark blue curves: nullclines at the beginning of the parameter drift at $t = 0$ and $c = 0.4$, dark green curves: nullclines at the end of the parameter drift at $t = T_{end} = 200$ and $c = 0.01$. The non-autonomous basins of attraction are presented at the end of the simulation $T_{end}$ in light green, yellow, violet and light blue. c) For comparison: Basin of attraction in the frozen-in case for $c = 0.01$.

change but also depending on the timescale separation between the two subsystems (Fig. 10a,b). Comparing the slow and the fast drift of the environmental parameter, i.e., the coupling strength, we observe that even two basins have "disappeared". To compare with the frozen-in case we present also its basins onattraction in Fig. 10c.

When we continuously vary the rate of environmental change to identify the rate-dependent masking effect, we find that
the relative size of the non-autonomous basins of attraction and the tipping probabilities vary non-monotonously in the case of different timescales $\varepsilon = 0.1$ (Fig. 11a,b). This finding demonstrates that the masking effect for bifurcations in more general systems exhibits a highly complex dependence on the timescales, which makes it rather difficult to predict.

### 3.5 Coupled systems with diffusive coupling

Finally, we address the third coupling scheme, diffusive coupling, often used when coupling systems bidirectionally with the
same coupling strength $c_1 = c_2 = c$. In this case, the overall effect of the coupling depends on the difference between the variables. The corresponding model system reads:

$$
\begin{aligned}
\dot{X}_1 &= -(X_1 - s1)(X_1 - s2)(X_1 - s3) + c(X_2 - X_1), \\
\dot{X}_2 &= -\varepsilon(X_2 - s1)(X_2 - s2)(X_2 - s3) + c(X_1 - X_2), \\
\dot{c}_{1,2} &= \begin{cases} v & \text{for } 0 \leq t \leq T_r, \\ 0 & \text{for } T_r < t \leq T_{end}. \end{cases}
\end{aligned} \tag{11}
$$

For ecological systems, this would be the appropriate coupling when considering two populations in two different habitats coupled through species migration. The same coupling would be used for coupled chemical systems, where diffusion is assumed to be the most important spatial transport process. Following the same protocol of numerical simulations with the same

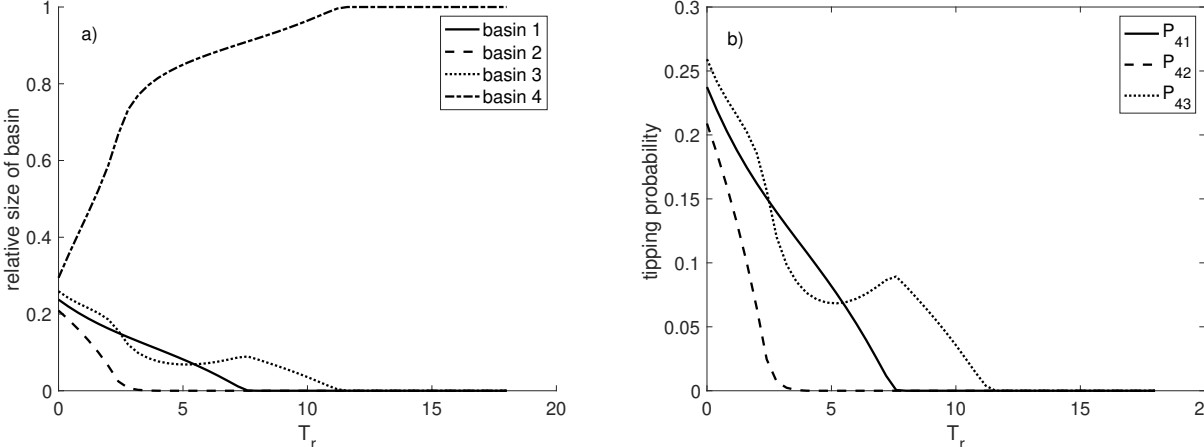

**Figure 11.** The relative size of the non-autonomous basins of attraction (a) and tipping probabilities (b) for the mutually coupled systems depending on the rate of environmental change measured in terms of $T_r$ ( $\varepsilon = 0.1$, $c_{start} = 0.4$, $c_{end} = 0.01$).

parameter values, we observe qualitatively the same behavior as for the other coupling schemes with one important difference. The masking effect occurs for much larger $T_r$, i.e., a much slower rate of environmental change, if the two systems are iden-
tical and no timescale separation occurs. This is due to the fact, that the effective coupling is much smaller than in cases of a master-slave coupling or a mutual coupling. The coupling strength is multiplied by the difference of the two variables in system 1 and 2 instead of the variable itself and this diminishes the effect of the coupling. Therefore, the overall effect for identical systems is much smaller than in the other cases (Fig. 12a,b). By contrast, if we introduce the timescale separation shown in Fig. 12c,d, we note, that the masking effect is again much stronger and comparable with the two other coupling but
now essentially determined by the timescale separation between the two systems 1 and 2. This emphasizes again the role of the timescale separation between coupled systems.

## 4  Discussion and Conclusions

We aimed to evaluate the consequences of a time-dependent variation of parameters or external forcing following a prescribed trend in a multistable system. In contrast to many other studies, our focus was not on the stable long-term behavior, i.e., the
attractors, but on the unstable sets of saddle type since their stable manifolds make up the basin boundaries. Specifically, we were interested in how the relative size of the basins of attraction varies in a non-autonomous system and how the "movement" of the corresponding basin boundaries influences the trajectories in state space. As already known from earlier works on rate-induced tipping, the time-dependent forcing implies that attractors like, e.g., stationary points become quasi-stationary and "move" through the state space according to the trend (Wieczorek et al., 2011; Ashwin et al., 2012). Whether a trajectory tracks
those quasi-stationary points or tips is the central question of rate-induced tipping. However, this property of quasi-stationarity

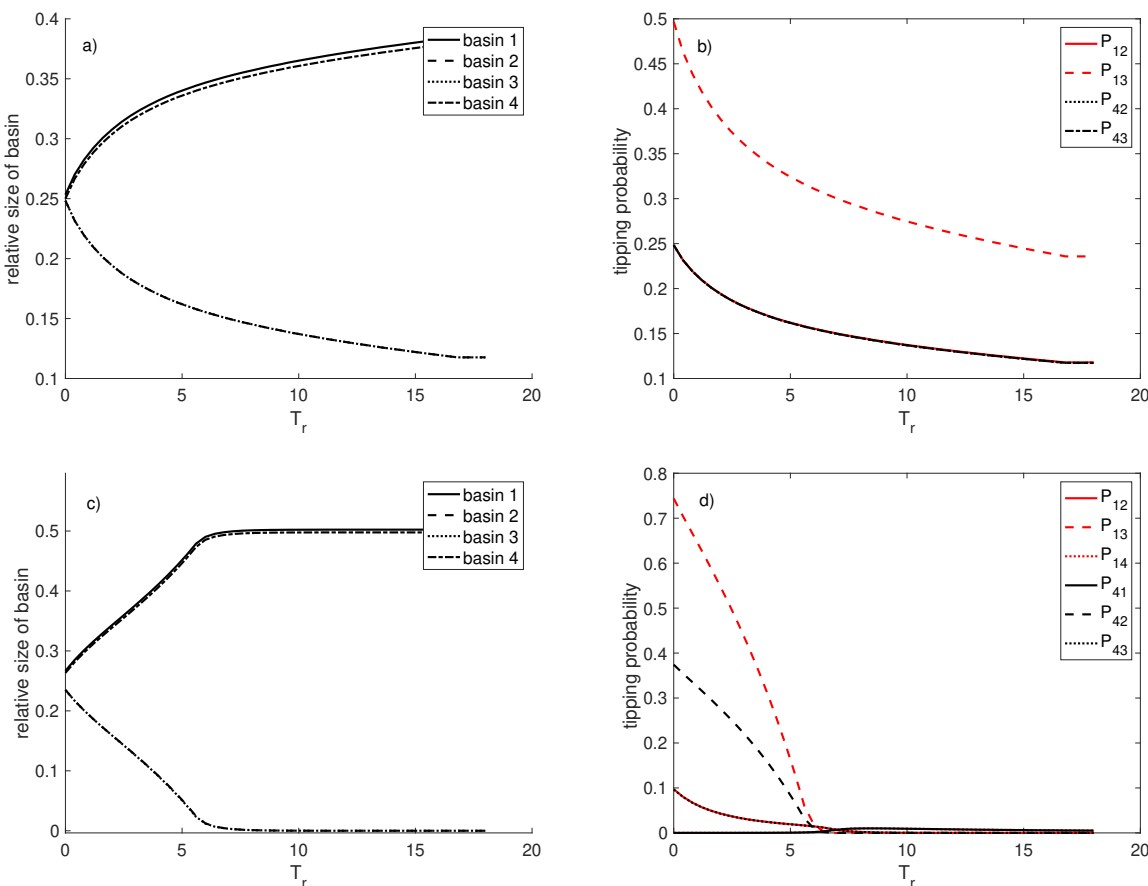

**Figure 12.** The relative size of the non-autonomous basins of attraction [a) and c)] and tipping probabilities [b) and d)] for the diffusively coupled system Eqs. (12) depending on the rate of environmental change measured in terms of $T_r$ ($c_{start} = 0.4$, $c_{end} = 0.01$) a) and b): $\varepsilon = 1$ and c) and d) $\varepsilon = 0.1$.

applies not only to the stable attractors but also to the unstable saddle points in the system. In a multistable system, in which several attractors coexist, those saddles are important determinants of the global dynamics since their stable manifolds make up the boundaries of the basins of attraction which organize a "partitioning" of the state space in the sense that they separate regions in state space with different qualitative behavior. Since the quasi-stationary saddle points "move" through the state space, the associate basin boundaries do that too, and this has tremendous consequences for trajectories as they can cross in non-autonomous systems the basin boundaries.

We have demonstrated the mechanism of basin boundary crossing employing a system from population dynamics possessing an Allee effect. In this model, the basin boundary crossing occurs for varying the critical population density corresponding to the "moving" saddle point making up the basin boundary. In the case of crossing, the trajectory tips when it meets the moving saddle point, i.e., the moving basin boundary. Then we addressed the question of what happens in higher dimensions when the

basin boundaries are not just saddle points but hypersurfaces in state space that are moving and/or even changing their shape. In addition, basins of attraction can even appear and disappear in bifurcations, e.g., in a saddle-node bifurcation. The classical computation of the relative size of a basin of attraction applies only to the frozen-in case with fixed parameters. To extend this approach to non-autonomous systems, we have called a non-autonomous basin of attraction the union of all those initial conditions which converge to a particular quasi-stationary state including the parameter drift. As an example, we analyzed two coupled, rather general bistable systems with different coupling schemes, which have been studied in a similar form already in the context of tipping cascades. In climate science, the most interesting coupling is the master-slave coupling, which is particularly used to investigate tipping cascades in two or a few coupled natural systems (Klose et al., 2020; Wunderling et al., 2021; Kroenke et al., 2020). One example is the impact of the melting of the Greenland ice sheet on the AMOC (Mehling et al., 2022; Klose et al., 2023) or tipping in unidirectionally coupled networks (Kroenke et al., 2020). Additionally, we have studied mutually coupled systems where each subsystem is a driver for the other. Finally, we have compared the results to a diffusive coupling used in many physical and ecological systems.

To study the impact of moving non-autonomous basin boundaries in detail, we have focused on a simple system with smooth basin boundaries in the whole parameter range, not fractal ones. The most straightforward situation in which basin boundaries are important is the saddle-node bifurcation of steady states in which a new stable steady state occurs together with a saddle point whose stable manifolds make up the basin boundary for the newly appearing steady state. Apart from details that are related to the different coupling schemes, the main findings are qualitatively the same for all of them. In our setup, two different timescales are involved, the intrinsic dissipative timescale of the dynamics of each subsystem and the timescale of the environmental change, which in our case was influencing only the coupling strength. We found that the relative size and the shape of the non-autonomous basins of attraction depend strongly on the rate of environmental change of parameters or external forcing. As a consequence, initial conditions that would converge to one attractor in the frozen-in case tip into another basin of attraction during the parameter drift. This leads to a partial tipping of trajectories. In addition, we showed that for each finally tipping initial condition, there exists a critical rate of environmental change for which this tipping occurs for the first time. Hence this tipping process fulfills the requirements of rate-induced tipping as defined in (Ashwin et al., 2012; O'Keeffe and Wieczorek, 2020). When the rate of environmental change becomes slower and slower, the newly appearing basins of attraction are less and less detected by the trajectories leading to a shrinking of the corresponding non-autonomous basins of attraction. This process happens gradually until even the last initial condition fails to tip to the newly formed basin beyond the saddle-node bifurcation. In other words, the corresponding basin of attraction "disappears" and the emergence of a new attractor is *masked* by the parameter drift.

The tremendous consequences of that masking effect become more evident when we discuss it from the point of view that anthropogenic changes in parameters/forcing are accelerating corresponding to an increasing rate, which means lowering the time interval $T_r$ (reading Figs. 7,11,12 from the right): Suppose our simulated trajectories would be observations starting at some time in the past to monitor the impact of changes in the environment. Assume further that the saddle-node bifurcation occurring along the change of environmental forcing gives rise to a new dangerous, possibly undesired stable state, implying dramatic changes in the global dynamics. Then we observe the following: the *slower* the environmental changes happen, the

smaller is the probability of detecting the change in the global dynamics in any of those observed time series. A critical rate of change exists below which the emergence of the coexisting dangerous state (beyond the saddle-node bifurcation) cannot be detected at all since none of the considered trajectories tips, i.e., the bifurcation, is masked. This *detection threshold* corresponds to the critical rate where the tipping probability to the newly formed dangerous state becomes, for the first time, positive. The tipping probabilities increase gradually and become considerably larger only when the environmental changes are already quite fast. This detection threshold beyond which the global change in the dynamics will be signaled by tipping trajectories depends strongly on the relationship between the timescale of environmental change and the intrinsic timescale of the system dynamics. In general, our results suggest that the faster the intrinsic timescales are compared to the rate of environmental change, the lower the probability of detection due to the small tipping probabilities. While for the master-slave and the mutual coupling, those detection thresholds occur for rather small $T_r$, i.e., rather fast critical rates, this threshold corresponds to a considerably larger $T_r$, i.e., a much slower rate of environmental change, for a diffusive coupling. However, the larger the timescale separation between the subsystems (here system 1 and 2), the faster becomes the critical rate for the detection of bifurcations. For all couplings holds: As long as the global change in state space remains masked or hidden, we would not detect the emergence of a new dangerous state by our monitoring time series starting in the past. Despite the fact that the new dangerous state has appeared without notice, other tipping mechanisms like noise-induced or shock-induced tipping could tip the system into that undesired state without any warning.

We have further unraveled the mechanism of how the trajectory tips from one basin of attraction to the other by crossing the basin boundary. This rate-induced basin crossing happens at the saddle point; either the trajectory "meets" the saddle point directly or it approaches first the neighborhood of its moving stable manifold and travels along it until the saddle point is reached for the crossing.

An analysis of the relative size of the basins of attraction and the tipping probabilities in a highly multistable system with fractal basin boundaries has been provided by Kaszás et al. (2019). This latter study focused on the statistics of tipping probabilities and a phenomenological description of the transitions happening when the parameter drift covers not only a saddle-node bifurcation but also other bifurcations and even a chaotic region. Our results here and the ones obtained by Kaszás et al. (2016); Kaszás et al. (2019) lead to the same conclusion: While most literature on multistable systems focuses on the investigation of attractors, i.e., the stable long-term states of a system, and their bifurcations, this study suggests a necessary shift in the paradigm of analyzing nonlinear dynamical systems in climate science, ecology and beyond: unstable saddle-type states, either saddle steady states or saddle periodic orbits with their associated stable manifolds as well as chaotic saddles with their associated stable foliations, are of equal importance and need a lot more attention in future studies on non-autonomous dynamical systems. Moreover, rate-induced tipping phenomena, which are closely related to those saddles, have to be identified since their "movement" in state space under a parameter drift determines the fate of any trajectory. Those unstable saddles are the organizing centers of the basins of attraction in a multistable system and as such play a decisive role in rate-induced tipping by crossing basin boundaries.

*Code availability.* This paper contains numerical simulations obtained with freely available numerical integrators for ordinary differential
equations (Runge-Kutta integrator Dormand-Prince 5(4)) available here: http://www.unige.ch/hairer/software

*Data availability.* All data in this paper are produced numerically, equations and parameters are given in the text.

*Author contributions.* The idea, the numerical simulations and the writing of the manuscript was all done by Ulrike Feudel.

*Competing interests.* The author is a member of the editorial board of journal Nonlinear processes in Geophysics. The peer-review process
was guided by an independent editor, and the author have also no other competing interests to declare.

*Acknowledgements.* U. F. would like to thank Ann Kristin Klose and Johannes Lohmann for inspiring discussions, Marie Arnold for some
preliminary simulations and Everton Medeiros for a careful reading of the manuscript. Moreover, U.F. thanks one of the unkown review-
ers for his/her manifold suggestions to improve the figures. U. F. acknowledges support from the German Science Foundation (Deutsche
Forschungsgemeinschaft) under grant No 454054251 (FEU 359/22) and by the European Union's Horizon 2020 Research and Innovation
program under the Marie Sklodowska-Curie Action Innovative Training Networks Grant Agreement No. 956170 (CriticalEarth).

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
