# Peer review of "Rate-induced tipping in ecosystems and climate: the role of unstable states, basin boundaries and transient dynamics"

_Nonlinear Processes in Geophysics, 2023_

## Author Response (AR1)

Dear editor,

I would like to thank you for handling this manuscript and finding very careful reviewers. I am grateful to those reviewers for their positive reviews and their numerous suggestions to improve the manuscript. Reviewer 2 had additionally many constructive ideas how to improve the figures. I have taken almost all their suggestions into account and the new version of the manuscript contains most of the figures in a new style and many extended explanations of the findings. All corrections of my English have been taken into account too. I hope that the new version of the manuscript can now be accepted for publication. Please find below the point-by-point answers to all the reviewers comments and suggestions and how this has been incorporated in the manuscript.

Sincerely yours,

Ulrike Feudel

**Response to Reviewer 1:**

I would like to thank the reviewer for his/her careful reading of the manuscript and pointing out some interesting references to be added. I have taken almost all suggestions of the reviewer into account and hope that the manuscript is now acceptable. Here is a description to all the changes to improve the manuscript according to the suggestions of the reviewer.

*As a general comment I would suggest to stress a bit more the question of "predictability" of tippings, especially in relation with the nature of tipping phenomena, since it is a central issue in many natural system and in the description with deterministic-stochastic approaches of natural phenomena.*

I would like to thank the reviewer for the suggestion to additionally address the question of predictability in the review part of the manuscript. I have included a whole subsection devoted to this question at the end of Sec. 2. This subsection includes a short discussion of developments of methodology to identify early warning signals when approaching critical transitions. In addition, I shortly recall some approaches of how to predict critical transition

including ideas of how to prevent them. This paragraph reads:

**Prediction of tipping points and early warning signals** In the course of climate change it becomes more and more important to find appropriate methods to predict tipping points and to identify early warning signals. One method that has been developed in the physics and chemistry literature is critical slowing down (CSD) of the restoring forces when a transition is approached [1–4]. Resulting from these smaller restoring forces that bring the system back to its stable state after a perturbation, the response to inevitable noise is amplified, leading to a rising standard deviation [5, 6] and a lag-1 autocorrelation when approaching the bifurcation [7, 8]. These methods—critical slowing down and noise amplification—have become extremely popular over the last decade as possible early-warning signals. Besides those methods other statistical approaches have been developed to estimate how close we are to tipping points in the climate system and in ecology [9–12] or how probable noise- and rate-induced transitions are [13]. They have been used to estimate the proximity of several tipping points in climate such as e.g. the melting of the Greenland ice sheet [14], the collapse of the Atlantic Meridional Overturning [15, 16] or the loss of the Amazon rainforest [17]. Despite these various applications of early warning signals they have also been critically discussed from various perspectives [18–20].

*Furthermore, another important issue to be highlighted could be the role of processes operating at different scales in changing the topology and the geometry of fixed point and attractors, as well as, the role of symmetries and scale-invariance (turbulence is one of the possible examples). Some possible suggested (optional) references are highlighted below.*

I have added some suitable references suggested by the reviewer to emphasize more the impact of different scales on the topological structure in state space. This is now included in the introduction as follows:

These timescale separations lead to partly unexpected behaviors. Such scale dependence has been studied in the literature in different contexts such as e.g. climate sensitivity [21], tipping in excitable systems [22, 23], overshooting and reversing tippings [24, 25] and the topological structure of invariant sets in complex systems including their characteristics like fractal dimensions [26, 27].

I would like to thank the reviewer for the numerous corrections of English in the manuscript All the suggestions concerning notation and language have been taken into account. But there have been specific questions, which I answer again one-by-one:

*Lines 37-49: I would suggest to add some general references to the different outlined points.*

Since these are very general statements, I find it strange to assign them to some authors.

*Eq. (4): the role of a rate-dependent b is like the albedo feedback in energy-balance models. It would be interesting to mention, if this is the case, as a possible example in the context of the present paper.*

I would like to thank the reviewer for this interesting thought. The energy-balance model looks indeed quite similar. However, to see that we would have the same behavior here, would need a thorough investigation and this cannot be explained in a half-sentence. Therefore, I omitted it here, but will undertake this investigation later.

*Lines 273-275: this corresponds to a quadratic map in Eqs. 4. Does this mean that an attractor and/or a closed basin is missing/forbidden?*

After the saddle-node bifurcation only one attractor is indeed left and this is the attractor corresponding to extinction. To avoid here a bifurcation-induced tipping, the ramping interval has been restricted to a value of b below the saddle-node bifurcation.

*Lines 389-390: is this an effect of the variations of the nature of the basins corresponding to $X_1 = 3, X_2 = 3$*

I thank the reviewer for noticing that the explanation is not clear here. I have added it to the text now as follows: The reason for this behavior is twofold. One reason is the separation between the intrinsic dissipative timescale and the timescale of environmental change. The

non-autonomous basins of attraction are computed starting from a grid of initial conditions, but all of them have already converged to the neighborhood of the attractor at $X_1 \simeq 3$, $X_2 = 3$, when the attractor $(X_1 \simeq 1, X_2 = 3)$ appears in the saddle-node bifurcation due to a faster intrinsic timescale. On the other hand, the example studied describes either chemical concentrations or abundances of species, which have to be positive or equal to zero. This restriction of possible valid initial conditions are another reason, why none of the used initial conditions can reach the new attractor. They can only do so in the frozen-in case, but not beyond a critical rate of environmental change.

*Section 3.3: it is clear that "drastic" effects are more evident for changing rate of environmental than timescale of the process, this is crucial for predictability and particularly true for climate change. A few explanatory/additional lines on this would be desirable.*

This explanation has been added as follows: Overall, we can conclude, that bifurcations which change the topological structure of the state space have a tremendous impact on the evolution of trajectories. This impact depends crucially on the relation between the intrinsic dissipative timescale and the timescale of environmental change. It turned out, that following the trajectories, which can be considered as observables do not necessarily detect those transitions. There is a detection limit beyond which bifurcations which happen in state space are not noticed by the observables. As a consequence transitions to those undetected states can happen without warning.

In addition I have extended the subsection on the diffusive couplingwhere the overall effect of the masking effect happens at much slower rates of environmental change. However, I have now also shown that those critical rates change change drmatically, when considering a larger timescale separation between the two coupled subsystems.

*Line 551: Code availability the link seems not to work.'*

Now the link should work, I corrected the mistake.

**Response to reviewer 2**

I would like to thank the reviewer for the very careful reading and the numerous suggestions to improve the manuscript. He/she has also made many suggestions to improve the figures which I have also taken into account. According to the suggestions of the reviewer I extended many of the explanations. I hope that the manuscript is now much better readable. I will now answer all the concerns of the reviewer point-by-point except for the corrections in English language, which I have all taken into account.

*The paper explains tipping phenomena so clearly that it could be a chance to also explain a few necessary basics of dynamical systems theory, in particular, what a "manifold" and "saddle points" are (terms are used a lot but not explained). This could make the paper more accessible for a broader readership.*

I would like to thank the reviewer for this suggestion. Now I included the basics on saddle points and their stable and unstable manifolds. These basics are now explained as follows:
The unstable state (red ball) located on the hill of the stability landscape marks the basin boundary. This boundary separates the two basins of attraction, i.e., the two set of initial conditions which all converge to one of the respective attractors. In higher dimensional systems these unstable states on the boundary are of saddle type, possessing stable and unstable manifolds. The stable manifolds are hypersurfaces in state space whose dimension is equal to the number of stable directions or stable eigenvalues of the corresponding Jacobian matrix of the saddle, while the unstable manifolds correspond to hypersurfaces determined by the number of unstable directions or eigenvalues. In the special case of a two-dimensional system the saddle steady state has two eigenvalues one stable and one unstable and the corresponding stable and unstable manifolds are one-dimensional. The stable manifolds along which trajectories move towards the saddle make up the basin boundaries.

In addition to this explanation I have also included an additional figure as Fig. 1b showing the saddle point and its stable manifolds as the boundaries of the basin of attraction.

*The manuscript cites the relevant literature as far as I can judge. There is a new study by*

*Ritchie et al. about a very similar topic which could be cited: https://journals.aps.org/pre/abstract/10.1103/*

I am grateful to the reviwer for pointing out this interesting reference, which I have now also cited in the manuscript.

*Title: Why "in ecosystems and climate"? The systems presented are so general that they are not restricted to these fields. Environmental change is of course a good context and illustrates the relevance of non-autonomous tipping, so the examples in the text are useful, but I am sceptical if the title should make the scope more narrow than it actually is.*

The reviewer is correct in saying that the material presented is much more general than the title says. The title is chosen according to the readership of this journal, which are in general scientists from geosciences. It was my intention from the very beginning to explain rate-induced transitions to a broad audience of geophysicists. Therefore all examples are taken from climate and ecology. Moreover, this paper is an invited contribution and the editors had precise ideas what the manuscript should talk about. But we are working already on a more general review for the broad readership of physicists, which will be submitted elsewhere.

*Comments about all figures showing the state space (X1 versus X2), i.e., Fig. 5, 6, 8, 9, 10*

I would like to thank the reviewer for the suggestions made to improve the figures. All figures are completely new and I took almost all the suggestions into account, in particular the size of the labels, the number of trajectories shown, the color coding, the size of the symbols of steady states, using a) and b) instead of left right and including arrows in the trajectories. I also changed completely the captions to explain all things shown in the figure. However, I could not realize some of the suggestions. These are: It is impossible to include the whole vector field since it would only represent the vector field at one particular time instant, but I wanted to show the entire time evolution. This works best with showing the whole time evolution of trajectories including the parameter drift. To avoid too much confusion, I reduced the number of trajectories to 4 'and omitted the indication of the different time intervals which the trajectories go through (to the first and the second

saddle-node bifurcation and the stop of the drift, which was indicated by dashed, dotted and white lines). It was meant to illustrate how far the trajectory got in state space, when the bifurcations occur. But I agree with the reviewer, that the most important parts, the dotted and white lines were barely visible so that the original purpose was not well illustrated. Now all trajectories are shown as full black lines. I have now indicated the starting point of the trajectories and included arrows in the trajectories. I changed the colors of the nullclines from candy-colors to darker ones for nullclines and to lighter ones for basins of attraction. But I did not change the colors in different panels. The colors represent at which time instant the nullclines are plotted, since they are all time-dependent, one color corresponds always to the starting point $t = 0$, the other to the end point $t = Tend$. This was only explained in the text, but I included it now also in the caption to make that very clear. The color coding of the non-autonomous basins of attraction is now also mentioned in the caption.

*These disturbances correspond to the displacement of the state from the valley of the fixed stability landscape, depicted as a vertical path of perturbations in Fig. 2. On the other hand, disturbances in the system parameters or external forcings change the stability landscape. Both types of disturbances are possible and have very different effects (Schoenmakers and Feudel, 2021)." Is this always clearly separable? For example, perturbing water density (state variable) and applying freshwater flux (forcing) in an ocean model could be argued to be essentially the same thing.*

Of course not and I mentioned that already in the text in the next sentence line 104-106. But I cannot follow fully the argumentation of the reviewer, because in his example one can separate the two, since changing the freshwater flux is the cause, but the change of the density is the response. Following the reviewer, one can also say that a perturbation of density (e.g. by noise) could be interpreted as fluctuations of the freshwater flux, which would also be a possible interpretation. However, I think that this discussion could also be very misleading since it is closely related to the example and could lead to confusion of the reader. Therefore, I omitted it in the text.

*line 125: "flow patterns in the ocean" ⇒ ocean circulation would probably be the more typical expression.*

I changed it in the text, since the cited papers indeed only include ocean circulation. However, "flow patterns" is a common expression for patterns in hydrodynamic flows.

*line 127-128: "In those systems, bi- or even multistability has been discovered". Could be misleading. There are some (often simple) models that show multistability, but complex ESMs usually do not, and there are large uncertainties. line 127-128, line 133-134: "even multistable" Strictly speaking, "multistability" includes bistability (so "or even" is not really adequate). line 130-131: "The other stable state would be a reverse circulation pattern." This may be the case in very simple models like Stommel's model, but as far as I know not in higher-complexity models. There, the alternative state is an "off" state, often with still some (weak) overturning, but no flow reversal. I suggest the author checks and adjusts this.*

I agree with the reviewer, that in this part the formulations were a bit sloppy. All facts the reviewer mentions are of course clear to me. Indeed, bistability is part of multistability. I wanted to make a distinction between bistability and "true" multistability with more than 2 attractors coexisting on purpose, since the latter is related to much more complex behavior and is much less understood. I have changed the text to highlight this difference in another way and added a very recent paper on that by Lohmann et al. 2023.
I have rewritten this part completely and it reads now: In those systems, bi- or even multistability, i.e. the coexistence of more than 2 stable states for the same environmental conditions, has been discovered. For the AMOC, mostly two different stable flow patterns exist: one of them can be considered as a conveyor belt transporting heat to the Northern latitudes, releasing this heat to the atmosphere, forming North Atlantic Deep Water (NADW) which is transported back to the Southern latitudes at considerable depth. This would be the state, where the heat transfer to the north in "on". The other stable state is related to an "off" state. This bistability can give rise to a possible breakdown of the AMOC, which has been discussed employing several conceptual models [28–31]. In those conceptual models often the second state is related to a reverse circulation. In large ocean circulation models this bistability has also been confirmed [32], with an "off" state which does not relate to a reverse circulation but to a very weak circulation northwards. In large ocean circulation models it has been shown, that the system can even exhibit the coexistence

of several different flow patterns related to different spatial patterns of heat transfer to the atmosphere [33]. This occurrence of multistability has been confirmed recently with other high-resolution models [34, 35].

*140-141: "As a result, the system tips or collapses from a coral-dominated into an algae-dominated reef (Holbrook et al., 2016)." Again, my impression is that there is much more uncertainty than this sentence suggests. I'm not so familiar with this research field, but I believe that the question whether coral reefs display several stable states is inconclusive.*

I agree with the reviewer that one should formulate more carefully, though Holbrook et al. use the terminology of tipping points in their study. That the collapse occurs and what are the possible reasons for it, is widely accepted in the ecological literature based on observations. However, that this can be interpreted as a tipping in the dynamical systems sense, is questionable as the existence of thresholds in ecology in general is controversially debated in ecology. I have added a corresponding citation Hillebrand et al. 2021. Therefore, I cut the word "tips" in the sentence.

This part reads now: Examples of alternative states in ecosystems have been discussed in the literature (cf. [36] and references therein), though the existence of thresholds in ecology is controversely debated [37]. A prominent example in which such transitions from one stable state to another are nowadays already observed are tropical coral reefs, which are found to be overgrown with green algae due to climate change and other anthropogenic and non-anthropogenic influences. As a result, the system collapses and exhibits a shift from a coral-dominated into an algae-dominated reef [38].

*There is a confusion of terms here that unfortunately is very common in the "tipping points" / dynamical systems literature. A noise-induced transition refers to a (possibly radical) change in the pdf of the state when the noise intensity (a parameter of the stochastic system) is changed. This is also the definition used in Horsthemke and Lefever 1984, as well as Kuehn 2011, which the author cites here. An example of this phenomenon would be stochastic resonance (in case an oscillating forcing is also applied in addition). A noise-induced transition is therefore not the same as N-tipping (Ashwin et al., 2012), or "noise-induced tipping" as the author calls it here, which just describes a single event in*

*a single realisation of the system. It could be a good opportunity to clarify this common misunderstanding in the paper, or at least the paper should avoid the misleading term of "noise-induced transitions".*

I would like to thank the reviewer for pointing out this confusion in the two notations, what a noise-induced transition is. In the literature indeed different descriptions are used for the phenomenon called "noise-induced transition". To outline this a bit further I have changed the text accordingly and also cite again Ashwin whose definition of N-tipping is different.

The text reads now: Though this description sounds like a result of a single trajectory, it needs a stochastic description, since one has to study ensembles of trajectories with different realizations of the noise and probability distributions (pdfs) over the state space. There is a vast literature on noise-induced transitions in many different science disciplines and the notion of noise-induced transitions differs across the literature. While many studies classify noise-induced transitions as qualitative changes in the aforementioned pdf with the noise strength as the bifurcation parameter [39, 40], other works focus directly on the transition from one stable state to another mentioned under the influence of noise as outlined above. Ashwin et al. [41] define N-tipping as a system which leaves the neighborhood of a quasi-stationary state due to the influence of fluctuations.

*line 161: "…living in the sediment of the North Sea at the end of the 90s" is a bit confusing (order of words). The 90ies refers to the regime shift mentioned earlier I suppose.*

This is a mistake in English, which I have corrected now. Indeed, the regime shift has been observed in the 90s.

*Fig. 3: The cyan colour is hard to see; it is unclear how the potential is accelerated (to what side) – to the left according to the trajectory of the ball, but the arrows point to the right; it is not visible where the ball would cross an unstable equilibrium in case e (see point about absence of basin-crossing above). Fig. 3 and line 204-214: I am a bit confused why a large excursion in phase space without crossing any equilibrium point counts (no basin crossing) as a tipping here. Isn't this definition a bit arbitrary and too qualitative? If the ball*

*never leaves its basin and does not even cross an unstable equilibrium (or basin boundary),
what excursion is large enough to constitute a tipping?*

On purpose I have not provided an illustration for the basin crossing, since this is discussed later in the manuscript in more detail. Fig.3 is meant to illustrate the other case of rate-induced tipping which is not related to a basin boundary crossing. It is related to the mechanism discussed in Wieczorek et al. 2011 and Vanselow et al. 2019, where the rate-induced tipping occurs without the existence of an alternative state. It occurs in slow-fast systems and requires a bended critical manifold possessing a fold. The exact mathematical conditions are outlined in Wieczorek et al. 2011. I have extended the explanation of this part of Fig. 3 to make this more clear.

It reads now: Since all aforementioned tipping mechanisms are related to the coexistence of alternative states, rate-induced tipping can also occur when there is only one stable state present and the system is characterized by different timescales (slow-fast system). The dynamics of such systems can be described by so called critical manifolds in case of a perfect timescale separation or slow manifolds, when the timescale separation is finite. In case of a complex structure of the critical manifolds, for instance when these manifolds have stable and unstable parts which meet in a fold, then a rate-induced crossing of this fold can make the trajectory visit very different parts of the state space far away from the original stable state and perhaps even dangerous for the system. This mechanism of rate-induced critical transitions is illustrated in Fig. 3d,e where the whole stability landscape is moved at a certain rate. Suppose that the stability landscape in Fig. 3d is pulled with a certain rate towards the observer. Consequently, the ball will no longer be located in the minimum of the valley but will be displaced to the left. The restoring forces will start acting, and the ball begins to "roll" to catch the moving minimum. If the pulling rate is slow, then the stable state (ball) follows, or we say it *tracks* the minimum of the stability landscape. By contrast, in Fig. 3e, the rate of "pulling away" the stability landscape is much faster or comparable with the timescale of the restoring forces. In this case, the ball lands in a completely different region in state space, leaving the minimum's proximity and leading to qualitatively different dynamics. This large excursion in state space corresponds to rate-induced tipping since the system visits very different parts of the state space with qualitatively different behavior. If the change in the environmental conditions stops, this visit to a different state will be

transient and, finally, the system returns to the stable quasi-stationary state, which has moved. This transient dynamics could lead to qualitatively different states, like population collapse in predator-prey systems [42] or population outbreaks [43].

*line 224-232: I find the case of spatially extended systems particularly interesting. These systems are often neglected in other "tipping point" related literature, so I here see the opportunity that this paper could add something. Some more text about this case with an example might thus be a good idea. This could be merged with Sect. 3.5 which treats diffusive systems (but also too briefly).*

I have extended this paragraph with an example according to the suggestion of the reviewer. A merging with Sect. 3.5 does not fit, since the description of tipping in spatially extended systems deals with tipping in partial differential equations and in particular with pattern formation, while Sect. 3.5 deals with a simplified version of a kind of network approach with just two nodes. The extended paragraph reads now: Several examples have been studied by Meron and coworkers in dryland vegetation models [44–46]. These models of different complexity study the interplay between vegetation and soil water. Besides the homogeneous states "bare soil" and "full vegetation cover" there exist depending on the environmental conditions, in general the precipitation level, different patterns like holes in the vegetation, stripes and spots of vegetation. These patterns can coexist and fronts separating the different pattern can occur. The speed of the fronts determines the speed with which one patterns is exchanged by the other leading to a gradual tipping between different patterns in the whole area.

*Sect 3.1: A figure with the equilibria of the population model would be a nice addition to the equations, maybe as part of Fig. 4.*

After rethinking this suggestion I do not see any need for another figure containing the steady states. The state space is one dimensional, all steady states are located in one line which is already visible in Fig. 4 at the ordinate. Therefore I omitted this figure.

*Caption of Fig. 4: why does it refer to "Eqs. 5"? Isn't this system described by Eq. 4?*

I would like to thank the reviewer for pointing out this mistake. It has been corrected.

*lines 314-333: Fig. 5 is mentioned only very late, could refer to it as soon as the concept of nullclines is introduced; I think this would help to get the idea.*

The whole paragraph has been rearranged and reads now: Rescaling the time in terms of $\tau = \varepsilon_1 t$, it turns out that only the ratios between the intrinsic timescales of the different subsystems $\varepsilon_2/\varepsilon_1$ and the ratios between the timescale of transport or coupling and the intrinsic timescale (like, e.g., $c/\varepsilon_1$) are important. Therefore, we will continue the analysis with the rescaled equations:

$$\dot{X}_1 = -(X_1 - s_1)(X_1 - s_2)(X_1 - s_3) + cX_2, \tag{1}$$

$$\dot{X}_2 = -\varepsilon(X_2 - s_1)(X_2 - s_2)(X_2 - s_3), \tag{2}$$

where $\varepsilon$ and $c$ are the corresponding ratios.

System 2 appears as a driver or master for system 1. We analyze the dynamics in the most intuitive way, we use the concept of nullclines, which are given by the algebraic equations $\dot{X}_1 = f_1(X_1, X_2) = 0$ and $\dot{X}_2 = f_2(X_2) = 0$. While for the driver system 2, the nullclines are given by straight lines at the values of the three steady states of system 2, the nullcline of system 1 is represented by the cubic function $X_2 = (X_1 - s_1)(X_1 - s_2)(X_1 - s_3)/c$. The intersection points of $f_1 = 0$ and $f_2 = 0$ are the steady states of the master-slave system. Their stability can be computed from the eigenvalues of the corresponding Jacobian. An illustration of two possible situation is given in Fig. 5a,b. Depending on the internal parameters $\varepsilon$ and the coupling strength $c$, the system possesses two, three, or four stable, steady states in the considered parameter range of coupling strength $c$. The shown two cases serve as the beginning [Fig. 5a] and the end [Fig. 5b] point of the parameter drift along a linear ramp. We are fixing all parameters (frozen-in case) and compute the attractors and their corresponding basins of attraction by choosing a grid of initial conditions in a specified region of state space and integrating them all in parallel until they reach the attractor. This allows us to compute also the relative size of the basins of attraction $\mathcal{B}_\mathcal{A}$ as the quotient of the number of initial conditions converging to attractor $\mathcal{A}$ divided by the total number of initial conditions taken into account [47]. Fig. 5 shows that the state space

is "partitioned" into different basins of attraction indicated by different colors with basin boundaries separating them. In the frozen-in case, the basin boundaries are invariant sets that cannot be crossed by trajectories and, hence, represent rigid boundaries in state space for the trajectories.

*line 402 + Fig. 7b: It may be a bit misleading to refer to "tipping probabilities". This term makes me think of a stochastic system with always the same parameters and initial conditions but different noise realisations. e.g., see https://journals.aps.org/pre/abstract/10.1103/PhysRevE.95.05220 But here, the outcome depends only on the distribution of the initial conditions. I would rather call it "frequency" or "fraction" of tipping trajectories.*

Since I am using the same measure as introduced in Kaszás et al. 2019 I have kept the name and inserted also the formula to avoid confusion, though I agree with the reviewer that "fraction" would be more appropriate. I have inserted this now in the text: We follow here the approach introdcued in [48] and compute the tipping probabilities $\mathcal{P}\mathcal{A}_1 A_2$. We calculate first the relative size of the non-autonomous basin of attraction $\tilde{\mathcal{B}}(\tilde{A}_2)$ for the quasisteady state $\tilde{A}_2$ at the end of the parameter drift simulation $T_{end}$ and take its intersection with the frozen-in basin of attraction $\mathcal{B}$ of the frozen-in attractor $\mathcal{A}_1$ at the beginning of the simulation normalized by the frozen-in basin of attraction $\mathcal{B}(\mathcal{A}_1)$ at the beginning of the simulation. Loosely speaking, we calculate that fraction of initial conditions which would have converged to the attractor $\mathcal{A}_1$ in the frozen-in case, but which during the parameter drift tip to the basin of attraction of the moving quasisteady state $\tilde{A}_2$. In mathematical terms this can be expressed as follows:

$$\mathcal{P}\tilde{\mathcal{A}}_1 \tilde{A}_2 = \frac{\tilde{\mathcal{B}}(\tilde{A}_2) \bigcap \mathcal{B}(\mathcal{A}_1)}{\mathcal{B}(\mathcal{A}_1)} \tag{3}$$

The continuous change of the tipping probabilities shown in Fig. 8 indicates that we observe partial tipping, i.e. each trajectory possesses its own critical rate at which it tips.

*line 423-426: A bit short. Some more description and explanation in words about what differences we see between the plots, and why, would be great. Also, it's not clear to me why we need to epsilons. Could we not just remove epsilon 1 in this system? Eqs. 11-13. Again, why two epsilons? Would one not be enough to capture a time scale separation? As you*

*write yourself in lines 450-454, only the ratio matters.*

I have changed the whole notation in the new version of the manuscript and rescaled time by $\epsilon_1$ as discussed earlier in Sect. 3.4. This explanation in Sect. 4 is now removed, since the whole manuscript deals only with the ratios of the parameters.

*Fig. 10: It could help to have one figure showing the static system similar to Fig. 5 for the coupled system.*

This figure for the frozen-in basins is now added as Fig. 10c.

*You could call Eq. 8-10 system 1, and Eq. 11-13 system 2; it would help when refering to these. Equations 6-7 and 8-9 are identical; could remove one pair.*

since the names system 1 and system 2 was already given to the two coupled subsystems, this suggestion does not wotk out. But I have now added in all the captions the reference to the mathematical equations on which the shown pictures are based on.

*Fig. 12: Due to the overlap, it is hard to see which two basins change change together. One might use a combination of colour and line type to solve this.*

I tried it out, but it did not look any better, so I left the figures as they were. The line types are a combination of the two when overlapping, but I agree, it is hard to see.

*Sect 3.5 appears only as an afterthought, though diffusive coupling plays an important role in many models of many different systems. I suggest to expand this section, adding a similar analysis as for models 1 and 2 above, or at least explain in some more detail how and why the results differ.*

Since there is no qualitative difference between the results in Sec. 3.3 and 3.4 compared to Sec. 3.5. I expanded the section by an explanation why there are quantitative differences. Instead of presenting only the example with a timescale separation I have now added also

the case of no timescale separation to compare with.

Here is the explanation added: For ecological systems, this would be the appropriate coupling when considering two populations in two different habitats coupled through species migration. The same coupling would be used for coupled chemical systems, where diffusion is assumed to be the most important spatial transport process. Following the same protocol of numerical simulations with the same parameter values, we observe that qualitatively we observe the same behavior as for the other coupling schemes with one important difference. The masking effect occurs for much larger $T_r$, i.e., a much slower rate of environmental change, if the two systems are identical and no timescale separation occurs. This is due to the fact, that the effective coupling is much smaller than in cases of a master-slave coupling or a mutual coupling. The coupling strength is multiplied by the difference of the two variables in system 1 and 2 instead of the variable itself and this diminishes the effect of the coupling. Therefore, the overall effect for identical systems is much smaller than in the other cases (Fig. 12a,b. By contrast, if we introduce the timescale separation shown in Fig. 12c,d, we note, that the masking effect is again stronger now taken over by the timescale separation. This emphasizes again the role of the timescale separation between coupled systems.

*line 535-537: "either the trajectory "meets" the saddle point directly or it approaches first the neighborhood of its moving stable manifold and travels along it until the saddle point is reached for the crossing." - This is a very qualitative distinction, isn't it?*

I agree with the reviewer that this is just a qualitative description. To make this more quantitative and precise, other methods are needed than simulations. This research is already under way, but will be published elsewhere, since this still needs a thorough analysis.
* * *
[1] M. Heinrichs and F. Schneider, J. Phys. Chem. **85**, 2112 (1981).

[2] N. Ganapathisubramanian and K. Showalter, J. Phys. Chem. **87**, 1098 (1983).

[3] J. Tredicce, G. Lippi, P.Mandel, B. Charasse, A. Chevalier, and B. Picque, Am. J. Phys. **72**, 799 (2004).

[4] M. Scheffer, J. Bascompte, W. Brock, V. Brovkin, S. Carpenter, V. Dakos, H. Held, E. van

Nes, M. Rietkerk, and G. Sugihara, Nature **461**, 53 (2009).

[5] E. Surovyatkina, Nonlin. Processes Geophys. **12**, 25 (2005).

[6] S. Carpenter and W. A. Brock, Ecol. Lett. **9**, 311 (2006).

[7] H. Held and T. Kleinen, Geophys. Res. Lett. **31**, L23207 (2004).

[8] V. Dakos, M. Scheffer, E. van Nes, V. Brovkin, V. Petoukhov, and H. Held, Proc. Natl. Acad. Sci. USA **105**, 14308 (2008).

[9] T. M. Lenton, Nature Climate Change **1**, 201 (2011).

[10] T. M. Lenton, V. N. Livina, V. Dakos, E. H. van Nes, and M. Scheffer, Phil. Trans. R. Soc. A **370**, 1185 (2012).

[11] J. Fan, J. Meng, J. Ludescher, X. Chen, Y. Ashkenazy, J. Kurths, S. Havlin, and H. J. Schellnhuber, Phys. Rep. **896**, 1 (2021).

[12] J. J. Clarke, C. Huntingford, P. D. L. Ritchie, and P. M. Cox, Environmental Research Letters **18** (2023).

[13] P. Ritchie and J. Sieber, Phys. Rev. E **95** (2017).

[14] N. Boers and M. Rypdal, Proc Natl. Acad. Sci. USA **118** (2021).

[15] C. Boulton, L. Allison, and T. Lenton, Nature Commun. **5**, 5752 (2014).

[16] N. Boers, Nature Climate Change **11**, 680+ (2021).

[17] C. A. Boulton, T. M. Lenton, and N. Boers, Nature Climate Change **12**, 271+ (2022).

[18] P. Ditlevsen and S. Johnsen, Geophys. Res. Lett. **37**, 2 (2010).

[19] C. Boettiger and A. Hastings, Proc. R. Soc. B **279**, 4734 (2012).

[20] T. J. W. Wagner and I. Eisenman, Geophysical Research Letters **42**, 10333 (2015).

[21] R. Bastiaansen, A. Doelman, M. B. Eppinga, and M. Rietkerk, Ecology Letters **23**, 414 (2020).

[22] S. Pierini and M. Ghil, Sci. Rep. **11** (2021).

[23] A. Vanselow, L. Halekotte, P. Pal, S. Wieczorek, and U. Feudel, *Rate-induced tipping can trigger plankton blooms*, arXiv 2212.01244 (2022).

[24] P. D. L. Ritchie, J. J. Clarke, P. M. Cox, and C. Huntingford, Nature **592**, 517 (2021).

[25] P. D. L. Ritchie, H. Alkhayuon, P. M. Cox, and S. Wieczorek, Earth System Dynamics **14**, 669 (2023).

[26] T. Alberti, D. Faranda, V. Lucarini, R. Donner, B. Dubrulle, and F. Daviaud, CHAOS **33** (2023).

[27] G. D. Charo, M. D. Chekroun, D. Sciamarella, and M. Ghil, CHAOS **31** (2021).

[28] H. Stommel, Tellus **13**, 224 (1961).

[29] S. Rahmstorf, Climate Dynamics **12**, 799 (1996).

[30] C. Rooth, Progress in Oceanography **11**, 131 (1982).

[31] R. A. Wood, J. M. Rodriguez, R. S. Smith, L. C. Jackson, and E. Hawkins, Climate Dynamics **53**, 6815 (2019).

[32] W. Weijer, M. E. Maltrud, M. W. Hecht, H. A. Dijkstra, and M. A. Kliphuis, Geophysical Research Letters **39** (2012).

[33] S. Rahmstorf, Journal of Climate **8**, 3028 (1995).

[34] O. Mehling, K. Bellomo, M. Angeloni, C. Pasquero, and J. von Hardenberg, Climate Dynamics (2022).

[35] J. Lohmann, H. A. Dijkstra, M. Jochum, V. Lucarini, and P. D. Ditlevsen, *Multistability and intermediate tipping of the Atlantic ocean circulation*, arXiv 2304.05664 (2023).

[36] C. Folke, S. Carpenter, B. Walker, M. Scheffer, T. Elmqvist, L. Gunderson, and C. Holling, Annual Review of Ecology Evolution and Systematics **35**, 557 (2004).

[37] H. Hillebrand, I. Donohue, W. S. Harpole, D. Hodapp, M. Kucera, A. M. Lewandowska, J. Merder, J. M. Montoya, and J. A. Freund, NATURE ECOLOGY & EVOLUTION **4**, 1502+ (2020).

[38] S. J. Holbrook, R. J. Schmitt, T. C. Adam, and A. J. Brooks, Scientific Reports **6** (2016).

[39] W. Horsthemke and R. Lefever, *Noise-induced Transitions* (Springer, Berlin, 1984).

[40] C. Kuehn, Physica D **240**, 1020 (2011).

[41] P. Ashwin, S. Wieczorek, R. Vitolo, and P. Cox, Philosophical Transactions of the Royal Society A **370**, 1166 (2012).

[42] A. Vanselow, S. Wieczorek, and U. Feudel, Journal of Theoretical Biology **479**, 64 (2019).

[43] A. Vanselow, L. Halekotte, and U. Feudel, Theoretical Ecology **15**, 29 (2022).

[44] Y. Zelnik, S. Kinast, H. Yizhaq, G. Bel, and E. Meron, Phil. Trans. R. Soc. A **371**, 20120358 (2013).

[45] G. Bel, A. Hagberg, and E. Meron, Theoretical Ecology **5**, 591 (2012).

[46] Y. Zelnik, P. Gandhi, E. Knobloch, and E. Meron, Chaos (2018).

[47] U. Feudel, C. Grebogi, B. R. Hunt, and J. A. Yorke, Phys. Rev. E **54**, 71 (1996).

[48] B. Kaszás, U. Feudel, and T. Tél, Scientific Reports **9**, 8654 (2019).

---

## Author Response (AR2)

Dear editor,

thank you very much for the acceptance of my manuscript. In particular I would like to thank you for the very careful reading of the revised version and pointing out many typos and small incorrectnesses in language. I have incorporated all of those suggestions.

Sincerely yours,

Ulrike Feudel